# Mapping the Grounding Zone of Larsen C Ice Shelf, Antarctica, from ICESat-2 Laser Altimetry

Tian Li[1], Geoffrey J. Dawson[1], Stephen J. Chuter[1], Jonathan L. Bamber[1]

[1]Bristol Glaciology Centre, School of Geographical Sciences, University of Bristol, Bristol, BS8 1SS, UK

*Correspondence to*: Tian Li (tian.li@bristol.ac.uk)

**Abstract.** We present a new, fully automated method of mapping the Antarctic Ice Sheet's grounding zone using a repeat track analysis and crossover analysis of newly acquired ICESat-2 laser altimeter data. We map the position of the landward limit of tidal flexure and the inshore limit of hydrostatic equilibrium, as demonstrated over the mountainous and hitherto difficult to survey grounding zone of Larsen C Ice Shelf. Since the start of data acquisition in 2018, our method has already

achieved a near nine-fold increase in the number of grounding zone observations compared with ICESat, which operated between 2003 and 2009. We have improved coverage especially over the previously poorly mapped Bawden and Gipps Ice Rise and Hearst Island. Acting as a reliable proxy for the grounding line, which cannot be directly imaged by satellites, our ICESat-2-derived landward limit of tidal flexure locations agree well with independently obtained measurements, with a mean absolute difference and standard deviation of 0.39 km and 0.32 km, respectively, compared to interferometric synthetic

aperture radar-based observations. Our results demonstrate the efficiency, density and high spatial accuracy with which ICESat-2 can image complex grounding zones, and its clear potential for future mapping of the pan-ice sheet grounding zone.

## 1 Introduction

Long-term satellite observations have linked the on-going thinning of Antarctic ice shelves (Paolo et al., 2015) with

enhanced rates of ice discharge across the grounding line (hereinafter referred to as the GL) - the point where the grounded ice sheet first detaches from the bedrock and begins to float (Fricker and Padman, 2006). Ice discharge calculations are sensitive to the assumed location of the GL and therefore accurate GL mapping is required for mass balance estimates of the grounded ice sheet using the Input-Output Method (IOM) (Chuter and Bamber, 2015; Rignot et al., 2019). Changes in GL position are a key indicator of changes in the dynamical balance of the ice sheet (Schoof, 2007). Rapid GL retreat of glaciers

in the Amundsen Sea Sector between 1992 and 2011 observed from ERS-1/2 satellite radar interferometry (Rignot et al., 2014), and the accelerating retreat of ~65% of the GL along the Bellingshausen Sea Sector between 1990 and 2015 from Landsat optical images (Christie et al., 2016) reflect an ocean-driven glacial mass loss in West Antarctica. Thus, accurate knowledge of the GL position is critical for multiple applications including ice sheet numerical modelling (Joughin et al., 2010), mass budget studies (Shepherd et al., 2018) and assessing ice sheet stability (Favier et al., 2014).

The GL (Point G in Fig. 1) lies towards the landward edge of a transition zone between the fully grounded ice sheet (landward limit of tidal flexure shown as Point F in Fig. 1) and the freely-floating ice shelf (inshore limit of hydrostatic equilibrium shown as Point H in Fig. 1), forming the grounding zone (hereinafter referred to as the GZ). Within the GZ, there is Point I which is the break-in-slope where the surface slope changes most rapidly from the flat floating ice to the steep land ice (Fig. 1). While Point G cannot be detected directly from satellite-based observations, Point F lies close to this

location, and is thus generally considered to be the most robust satellite-observable proxy for Point G (Fricker and Padman, 2006; Fricker et al., 2009; Brunt et al., 2010b; Brunt et al., 2011; Rignot et al., 2011).

The most accurate method for estimating Point F is using differential synthetic aperture radar interferometry (DInSAR) (Rignot et al., 2011). This method, however, is constrained by the availability of suitable short temporal repeat pass SAR images, and there are relatively few regions where the method has been applied repeatedly (Friedl et al., 2019; Hogg et al.,

2017). It is also limited by the uncertainties in the external DEMs used in the geocoding of SAR images (Friedl et al., 2019; Milillo et al., 2017). Points F and H (Fig. 1) have previously also been derived using ICESat laser altimetry data (Fricker and Padman, 2006; Brunt et al., 2010b) and from pseudo crossovers of CryoSat-2 radar altimetry data (Dawson and Bamber, 2017; Dawson and Bamber, 2020). Both methods can provide additional GZ information across regions where the DInSAR-derived GZ information is unavailable. ICESat repeat track analysis proved to be a robust method for analysing the GZ, but

its coverage and temporal resolution were limited due to the requirement of multiple repeat tracks from different campaigns (Fricker and Padman, 2006; Brunt et al., 2010b). In addition, the approach used for ICESat-based GZ detection relied on visual interpretation, requiring a large amount of manual intervention (Fricker and Padman, 2006; Brunt et al., 2010b; Brunt et al., 2011). Point I mapped from satellite imagery has also been used to represent Point G (Bindschadler et al., 2011;Scambos et al., 2007), but this method often fails to identify fast-flowing glaciers with low surface slope where the

break-in-slope is less pronounced.

The launch of ICESat-2 on 15 September 2018, as a successor to the ICESat satellite mission, can achieve higher along-track sampling rate and better spatial coverage compared to its predecessor (Markus et al., 2017), providing the potential to map the GZ with greater accuracy and spatio-temporal coverage. ICESat-2 has a repeat cycle of 91 days. Compared to the single beam of the Geoscience Laser Altimeter System onboard ICESat, it measures the surface elevation of ice sheets using six

beams in three beam pairs emitted from the Advanced Topographic Laser Altimeter System (ATLAS) and enables an instantaneous determination of local across-track slope (Smith et al., 2019). The across-track spacing between each beam pair is approximately 3.3 km with a pair spacing of ~90 m. The along track sampling interval of each beam is 0.7 m with a nominal 17 m diameter footprint compared with the 170 m along-track spacing of ICESat (Markus et al., 2017). In this study, we investigate the ability of using ICESat-2 data to map the Antarctic GZ. We present a computationally efficient

technique to measure Points F and H by analysing repeat track data and crossover data from ascending and descending tracks. We chose Larsen C Ice Shelf in the Antarctic Peninsula to test this new methodology. It is one of the northernmost

ice shelves in Antarctica with widest across-track spacing and a range of different surface slopes at the GZ with complex topography inland (Jansen et al., 2010). It is, therefore, an ice shelf that presents a severe test for any new approach.

## 2 ICESat-2 Data

The first two cycles of ICESat-2 data acquired between 14 October 2018 and 30 March 2019 are not repeat cycles because the spacecraft pointing control was not yet optimized. In this study, we used 10 months of the ICESat-2 Land Ice Along-Track Height Product ATL06 version 3 data spanning from 30 March 2019 to 6 March 2020 (no data existed between 26 June 2019 to 26 July 2019 due to solar array anomaly of the satellite). Within this time period, there are four repeat cycles (3, 4, 5 and 6), however, part of cycle 4 (Reference Ground Tracks (RGT) numbered from 0 to 441) were missing and could not

be used in this analysis. The ATL06 elevation measurements are derived from the individual photon elevation observations, averaged over a 40 m length segment. The segments overlap by 50% along each of the six ground tracks, thus the ATL06 elevation measurements are separated by 20 m along each ground track (Smith et al., 2019).

The method used in this study of estimating GZ features relies on detecting the vertical movement of floating ice due to ocean tides. ICESat elevation data were routinely corrected for ocean and ocean loading tide and had to be 're-tided' (adding

tidal corrections back) (Fricker and Padman, 2006). While the ocean tide correction is not applied to the ICESat-2 ATL06 elevation (Neumann et al., 2019), we 're-tided' the ocean-loading tide in this study. We used the ATL06_quality_summary flag to remove poor quality elevation measurements which can be a result of clouds or random clustering of background photons (Smith et al., 2019). The along-track slope parameter was used to perform a height consistency check between adjacent elevation measurements along each ground track. This was achieved by calculating the neighbouring elevations

based on the along-track slope and comparing to the original surface elevations for the two neighbouring measurements (Arendt et al., 2019). We only used data where the differences between the original elevations and the estimated elevations were lower than 2 m. In addition, we also derived the locations of every reference segment for each ground track from the segment_quality group of the ATL06 product, which were used to calculate a reference track in the repeat track analysis in section 3. The ICESat-2 ATL06 elevation data from repeat cycles 3, 4, 5 and 6 on Larsen C Ice Shelf are shown in Fig. 2.

## 3 Methodology

### 3.1 Repeat track analysis

To identify the GZ features, we adopted a similar approach previously used with ICESat data by measuring the vertical motion of floating ice induced by ocean tides from a set of elevation anomalies for each repeat track (Brunt et al., 2010). The workflow of the repeat track analysis developed in this study includes four steps: 1) Repeat-track data preprocessing (section

3.1.1, box 1 in Fig. 3). 2) Elevation anomaly calculation (section 3.1.2, box 2 in Fig. 3). 3) GZ feature identification (section 3.1.3, box 3 in Fig. 3). 4) Filtering and visual validation (section 3.1.4).

### 3.1.1 Repeat-track data preprocessing

ICESat-2 generates six ground tracks in three beam pairs along one RGT (out of 1387 RGTs) in each repeat cycle (Fig. 1 in Smith et al. (2019)), forming six sets of repeat tracks. Figure 4a shows the four repeat tracks along the left beam in beam pair
1. Along one ground track, the across-track separation between different repeat tracks is approximately 10 m (Fig. 4a). Compared with ICESat data where the repeat track separation can exceed 100 m (Brunt et al., 2010b), this improvement can greatly reduce the errors in elevation change associated with across-track slopes.

For all six ground tracks numbered '1l' (left beam in beam pair 1), '1r', '2l', '2r', '3l', '3r', from different repeat cycles along each of the 1387 RGTs, we firstly obtained the elevation and geolocations of ATL06 measurements, as well as the
geolocations of reference segments. We then categorized the derived repeat tracks into distinct repeat-track data groups, each group was marked with a unique beam number and an RGT number. Only the repeat-track data groups (hereinafter referred to as 'single beam repeat-track data groups') with two or more repeat tracks were used for GZ calculation. For each single beam repeat-track data group, a 'nominal reference track' (black circles in Fig. 4a) at an along-track interval of 20 m was calculated by averaging the locations of reference segments from all repeat tracks. The use of reference segments to obtain
the nominal reference track can produce a common set of geolocation profiles free of the data loss of actual ground tracks.

Although we do not expect a substantial across-track slope over a ~10 m separation, elevation changes in some high sloping areas may still be affected by across-track slope. In order to reduce the errors in elevation change caused by large across-track slope and facilitate the automation of our repeat track analysis method, we generated three additional repeat-track data groups for each beam pair on top of the six single beam repeat-track data groups. A beam pair repeat-track data group was
created by merging the two single beam repeat-track data groups for the left beam and right beam in one beam pair. Each beam pair repeat-track data group was marked with a unique beam pair number ('1', '2', '3') and an RGT number. For example, the beam pair repeat-track data group shown in Fig. 4b includes all the repeat tracks along both the left beam and right beam in beam pair 1. As with the single beam repeat-track data group, a nominal reference track was calculated by averaging the locations of reference segments from all tracks inside the beam pair repeat-track data group, shown as the
black circles in the middle between the left and right ground tracks in Fig. 4b. The elevation of each track was then corrected for across-track slope onto this nominal reference track in section 3.1.2, this can reduce the errors associated with across-track slope. Altogether we have generated nine different repeat-track data groups along one RGT, six for single beam and three for beam pair. The GZ calculation was performed individually inside each repeat-track data group.

### 3.1.2 Elevation anomaly calculation

Before calculating the elevation anomalies for each track inside the repeat-track data group, a reference GL estimate is needed to define a GZ search window in the calculation (Fig. 3 box 2). For this purpose, we used the ESA Climate Change Initiative (CCI) (ESA Antarctic Ice Sheets CCI, 2017) grounding line derived from the DInSAR observations between 2015 and 2016 as this is the most up-to-date GL maps of Larsen C Ice Shelf. However, this product does not provide complete coverage, so we used the Depoorter et al. (2013) GL to fill in the data gaps and produced a composite GL for Larsen C Ice

Shelf. The Depoorter et al. (2013) GL is the most complete grounding line product to date and was compiled from a variety of GL datasets, including the break-in-slope points mapped from the Landsat-7 optical imagery in ASAID project between 1999-2003 (Bindschadler et al., 2011) and from the MODIS-based Mosaic of Antarctica between 2003-2004 (Scambos et al., 2007), DInSAR Point F from (Rignot et al., 2011) between 1994-2009, and the ICESat Point F between 2003-2008 (Brunt et al., 2010b; Brunt et al., 2011).

For each repeat-track data group, the reference GL was calculated as the intersection between the nominal reference track and the composite GL. Only ATL06 elevation measurements within a window size of 12 km landward and seaward of the reference GL along the nominal reference track were used. The grounding line of Larsen C Ice Shelf is stable and therefore unlikely to have experienced extensive changes (Konrad et al., 2018), thus the 24 km calculation window is suitable for GZ calculation in this region. Data points with the elevation higher than 300 m are most likely to be land ice as the surface

elevation of GZ is unlikely to exceed this threshold based on the hydrostatic equilibrium assumption. In order to improve the calculation efficiency, this part of the data was removed. The ICESat-2 elevation measurements located in open water were also discarded by using the coastline mask provided by the SCAR Antarctic Digital Database (ADD) (https://data.bas.ac.uk/items/862f7159-9e0d-46e2-9684-df1bf924dabc/#item-details-data).

        For repeat tracks in a single beam repeat-track data group, the average of elevations of each repeat track at the nominal

reference track was taken as the reference elevation for GZ identification. A set of 'elevation anomalies' were estimated by subtracting the reference elevation from the elevation profiles of each individual repeat track. The elevations and elevation anomalies of four repeat tracks for the right beam in track 1009 beam pair 2 are shown in Figs. 5a and 5b, respectively.

        For each track in a beam pair repeat-track data group, we firstly used the across-track slope parameter $\frac{dh}{dy}$ shown in Fig. 4b to correct the across-track slope induced elevation changes and interpolated to the nominal reference track. The across-track

slope $\frac{dh}{dy}$ for two ground tracks in a beam pair was calculated using the Eq. (1),

$$\frac{dh}{dy} = \frac{h_L - h_R}{y_L - y_R} \qquad\qquad (1)$$

where $h_L$, $h_R$ are the elevations of left and right ground tracks that make up a beam pair. $y_L$, $y_R$ are the across-track y-coordinates measured perpendicular to the RGT for the left and right ground tracks.

The interpolated elevation $h_{Ref}$ at the nominal reference track for the elevation of the left ground track, $h_L$, was calculated using the Eq. (2) (same for the elevation of the right ground track, $h_R$),

$$h_{Ref} = h_L - \frac{dh}{dy}(y_L - y_{Ref}) \tag{2}$$

where $y_{Ref}$ is the across-track y-coordinate measured perpendicular to the RGT for the nominal reference track (Fig. 4b). The average of all the elevations corrected for across-track slope from each track at the nominal reference track was taken as the reference elevation used in our identification of the grounding zone. After applying the across-track slope correction, a set of elevation anomalies were estimated by subtracting the reference elevation from the elevations corrected for across-track slope of each individual track.

### 3.1.3 GZ feature identification

Point F is identified as the point where the elevation anomaly of each repeat track first becomes significant, while Point H is defined where the elevation anomaly of each repeat track reaches its maximum and becomes consistent with the local tidal amplitudes (Fricker and Padman, 2006). Based on this definition, Points F and H were visually picked from ICESat repeat tracks by Fricker and Padman (2006) and Brunt et al. (2010b).

To automate the identification of Points F and H from the elevation anomalies, we first calculated the Mean Absolute Elevation Anomaly (MAEA) by averaging the absolute values of each elevation anomaly profile (Fig. 5c). We defined the region where the MAEA is close to zero (the region to the left of Point F in Fig. 1) as the fully grounded ice. Point F was then estimated to be the point where the gradient of the MAEA first increases from zero, and the second derivative of the MAEA reaches its positive peak. To reduce the influence of small-scale noise on the MAEA curve during the extraction of GZ features, a Butterworth low-pass filter with a normalized cut-off frequency of 0.032 and an order of 5 was applied (solid grey line in Fig. 5d). The low-pass filter removed the high-frequency noise without changing the shape of the MAEA curve. A set of multiple peaks were then extracted from the second derivative of the filtered MAEA. Despite the low-pass filter, noise still existed especially in the areas with complex topography, resulting in multiple peaks not associated with the GZ features. In order to select the correct peaks corresponding to the GZ features of interest, we fitted an Error function weighted by a Gaussian function with the variance of 0.005 to the MAEA (solid yellow line in Fig. 5d). The peak from the third derivative of the Error function at the landward region is a good measure of the approximate location of Point F, therefore, it was used as a guide point to select the correct GZ features. The closest positive peak (from the second derivative of filtered MAEA) to this guide point was identified as Point F. This allows the process to be automated in comparison to ICESat.

Similarly to the definition by Fricker and Padman (2006), we defined Point H as the point where the MAEA reaches a maximum and becomes stable, which is estimated to be the transition point where the gradient of the MAEA finally decreases to zero and the second derivative of the MAEA reaches its negative peak. Unlike the abrupt change in the gradient of the elevation anomaly at Point F, the gradient often tends slowly to zero at Point H (Brunt et al., 2010b; Dawson and Bamber, 2020). Therefore, using the same approach that we used for Point F, for Point H results in its location being slightly landward (vertical dashed grey line in Fig. 5d). Consequently, we used the peak from the fourth derivative of the Error function curve as the guide point. This point is closer to the transition point where the MAEA gradient finally decreases to zero, and the closest negative peak of the second derivative of filtered MAEA was used as point H. Additionally, the tidal height predictions at 5 km offshore from the reference GL for each repeat track were calculated from the CATS2008 Tidal Model, which is an update to the model described by Padman et al. (2002). The tidal heights provide an independent check for the tide-induced surface elevation changes at Point H. In order to assess the reliability of our GZ features in terms of the combined effect of tidal range and data coverage, the number of repeat cycles used in the GZ calculation and the mean tidal amplitude at Point H from the MAEA curve were also recorded.

### 3.1.4 Filtering and visual validation

Our algorithm is designed to take in both the single beam and beam pair repeat-track data groups as input, and produce a set of Points F and H in whatever conditions, to account for complex geographic features in different GZ regions. However, the GZ results can be filtered by applying a set of quality check flags.

The correct identification for GZ features depends on the accurate fitting of the Error function, which can be influenced by noise in elevation anomalies caused by across-track slope and small scale topographic features such as crevasses. Also the filtered bad elevation measurements due to snow and cloud in pre-filtering steps in section 2 can cause significant data gaps for some repeat-track data groups, making it impossible to identify the correct GZ features. To identify these failures in the approach, we applied two quality check flags. The first flag is data loss percentage. If the percentage of segments on the nominal reference track where there is no elevation measurement is larger than 50%, then it means this repeat-track data group does not have enough data to perform a reliable GZ calculation and the calculated GZ features were marked as 'Quality-1'. The second flag is inaccurate Error function fitting. Although noise in elevation anomalies related with across-track slope has been greatly reduced by using a single beam repeat-track data group over ~10 m separation and applying an across-track slope correction, the remaining across-track slope can still introduce errors for some repeat tracks. Together with crevasses, they can introduce a significant amount of noise in the MAEA curve and influence the final Error function fitting. The most prominent characteristic of this inaccurate Error function fitting, is the calculated Point F often locates several kilometres away from the reference GL. Here we calculated the distance between Point F and the reference GL, if it exceeded 5 km then this GZ was marked as 'Quality-2' to indicate potential Error function fitting problems. These two

quality flags highlight the majority of incorrectly identified GZ features. The remaining results which passed the quality checks were marked as 'Quality-0', indicating these GZ features are potential good results.

However, there are several circumstances where quality flags are inaccurate, so we performed a final visual validation on all the GZ results with the aid of the flags. For example, the GZ results marked with 'Quality-2' can be the existence of ice plain, which can result in up to 10 km separation between Point F and the reference GL (Brunt et al., 2011). Also, in category 'Quality-0', large across-track slopes can cause inaccurate GZ identification for single beam repeat-track data group (Figs. 6a-6d), but not for the beam pair repeat-track data group (which includes an across-track slope correction, Figs. 6e-
6h). To improve these results, we can manually set a new calculation window which only captures the ocean tidal signal. However, this requires more manual intervention, and does not significantly improve the coverage, as the nominal reference track of the beam pair repeat-track data group is only ~ 45 m away from either the left or right beam. Thus, inaccurate GZ data due to large across-track slope for single beam repeat-track data group were removed in the final visual validation.

**3.2 Crossover analysis**

Changes in ice shelf elevation due to tidal variation also can be calculated at the crossovers from ascending and descending tracks (Fig. 4c). Similarly to repeat track analysis, the elevation changes at crossovers on floating ice caused by tidal movement will be high, while they are close to zero on land ice where there is no vertical movement caused by ocean tides. Therefore, the elevation change from the crossover analysis can provide additional information on the GL location.

The crossover location was calculated by fitting latitude-longitude coordinates of all measurements from each ascending and
225 descending track into two quadratic functions and calculating the intersection of these two functions. Only data in proximity to the crossover location were used. We first found the two closest observations from the ascending track and the descending track close to the crossover location using a KDTree (k-dimensional tree) within a 100 m searching radius. We then extracted all the elevation measurements within a 100 m searching radius of these two closest data points from each ascending track and descending track, then calculated the actual location of crossover. The elevation at the crossover was
230 estimated by linearly interpolating the elevations from each ascending and descending track. If the ATL06 elevation measurements did not exist on both sides of the crossover within the 100 m searching distance, then the crossover of this track was discarded (Brenner et al., 2007).

The elevation change at the crossover not only includes the ocean tidal signal, but also a temporal signal of elevation change. In this study, we were only considering crossovers with a time difference less than 91 days to reduce the influence of
235 temporal elevation change. In addition, we deleted the crossovers where the elevation difference exceeded 10 m to remove other errors such as the geolocation errors of ICESat-2 (Smith et al., 2020). For crossovers on floating ice, if the time stamps of the ascending and descending tracks at the crossover are in the same phase of ocean tide cycle, the elevation change at this crossover should be close to zero, making it difficult to determine if the ice is floating or not. To eliminate these occurrences,

the CATS2008 Tidal Model (Padman et al., 2002) was used to calculate the tidal amplitude changes at each crossover and they were used as a reference for the vertical movement of floating ice. As the minimum detectable tidal amplitude from repeat track analysis is 20 cm after analysing all the GZ features calculated in section 3.1, we then set the minimum threshold of elevation change due to ocean tides on floating ice measured by the crossover analysis to be 40 cm by doubling the 20 cm threshold of repeat track analysis. If both the modelled tidal amplitude and the elevation change at crossover are lower than this threshold, the ascending and descending tracks are likely to be in the same tidal phase and this crossover was discarded. For crossovers calculated from different cycles at the same location, we also averaged the elevation differences at each crossover location.

## 4 Results and discussion

### 4.1 GZ distribution

Using the newly developed ICESat-2-based GZ detection algorithm in this study, we identified 253 Point F and 263 Point H over Larsen C Ice Shelf, which is a near nine-fold increase in the number of GZ features identified by ICESat (30 of each point (Brunt et al., 2010)). The spatial distribution of GZ features calculated from both ICESat and ICESat-2 are shown in Figs. 7a and 7b, together with a comparison of Point F determined from independent DInSAR observations from the ESA CCI product (ESA Antarctic Ice Sheets CCI, 2017) and Point H identified from Landsat-7 imagery in combination with ICESat data from ASAID project (Bindschadler et al., 2011) (Figs. 7c and 7d). The number of repeat cycles used to identify the GZ features from each repeat-track data group is shown in Fig. 7e and the mean tidal amplitude at ICESat-2-derived Point H is shown in Fig. 7f. The improvement in our ability to identify the GZ using ICESat-2 data is especially notable in heavily crevassed regions such as Jason Peninsula and Churchill Peninsula (Jansen et al., 2010), which were previously difficult to identify using ICESat observations alone. ICESat-2-derived Point F provides additional coverage in Hearst Island, Gipps and Bawden Ice Rise where ESA CCI DInSAR Point F is not available. Among these three regions, Gipps and Bawden Ice Rise are important pinning points for Larsen C Ice Shelf (Borstad et al., 2017).

Among all the ICESat-2-derived GZs, 162 Point F and 169 Point H were calculated from single beam repeat-track data groups, while 91 Point F and 94 Point H were calculated from beam pair repeat-track data groups. As the left and right beams are only separated by about 90 m and the GZ identified from the repeat track analysis for beam pair often locates in the middle between the left and right beams (~45 m in either direction), we do not expect there exist large deviations between these three GZs. In the same beam pair, we compared the locations of GZs calculated along the left beam and the right beam, and compared the GZ locations calculated from the beam pair repeat-track group and the two single beam repeat-track data groups. The mean absolute separations and standard deviations are shown in Table 1. The standard deviation between the locations of Point F at left and right beams is only 60.2 m, while the standard deviation of Point F between the single beam and the nominal reference track in the middle of the beam pair is 94.4 m. For Point H, the standard deviations

for these two comparisons are 349.3 m and 398.6 m, respectively. The similar magnitude in mean absolute separations and standard deviations between the GZs calculated from single beam repeat track analysis and beam pair repeat track analysis indicates these two methods having good internal consistency in identifying the GZs from ICESat-2.

Table 1. Mean absolute separations and standard deviations between the GZ features calculated from single beam repeat-track data group and beam pair repeat-track data group.

|  | Point F | | Point H | |
| --- | --- | --- | --- | --- |
|  | Mean absolute separation (m) | Standard deviation (m) | Mean absolute separation (m) | Standard deviation (m) |
| Left beam *vs* Right beam | 156.5 | 60.2 | 382.2 | 349.3 |
| Single beam *vs* Beam pair | 141.4 | 94.4 | 340.9 | 398.6 |

**4.2 Comparison with other GZ products**

The locations of Point F calculated from ICESat-2 data are in good agreement with the ESA CCI DInSAR Point F (Fig. 7a). The absolute mean separation (which was measured as the perpendicular distance from ICESat-2 Point F to DInSAR Point F line segment), and the standard deviation between ICESat-2 Point F and ESA CCI DInSAR Point F are 0.39 km and 0.32 km, respectively (Table 2). In comparison, the standard deviation between CryoSat-2-derived Point F and the ESA CCI
DInSAR Point F is 1.2 km in the same region (Dawson and Bamber, 2020). The absolute separations between the ESA CCI DInSAR Point F and ICESat-2 Point F are shown in Fig. 7c, where 65% of ICESat-2-derived Point F are located less than 0.5 km away from the DInSAR Point F. In addition, we compared our ICESat-2-derived Point F with the break-in-slope point estimated from the Landsat-7 imagery in ASAID project (Bindschadler et al., 2011). The break-in-slope point identified from optical imagery is free from the typical geocoding errors of SAR images in steep terrain especially when
using a low quality DEM. Moreover, Larsen C Ice shelf is a relatively slow flowing region so the ASAID break-in-slope point is a good representation of the GL. The absolute mean separation and standard deviation between these two products are 0.34 km and 0.28 km, respectively. The overall separations in Churchill and Kenyon Peninsula are smaller than the ESA CCI DInSAR Point F (Fig. A1).

ICESat-2-derived Points F located inside the inlet at Churchill Peninsula have the largest deviations from ESA CCI DInSAR
Point F (Figs. 7a and 7c), with an average of 2.78 km. The differences in position are not because of the incorrect interpretation of Point F from ICESat-2 data but likely due to the existence of a lightly grounded ice plain with low surface slope. Repeat track analysis for the two right beams from cycles 3 and 4 along track 506 beam pair 1 in this region (Fig. 2) is shown in Fig. 8. We manually defined the first break-in-slope of the surface elevation as the 'coupling point' in Fig. 8a (grey solid vertical line) (Corr et al., 2001). Between this coupling point and the ICESat-2-derived Point F, no tide-induced
elevation change is observed from the elevation anomalies in Fig. 8b. The elevation change signal around the coupling point

in the elevation anomalies are the errors associated with across-track slope. We interpreted the region between these two points as an ice plain according to Brunt et al. (2011). Point F can migrate several kilometres with ocean tides due to low surface slope inside the ice plain (Brunt et al., 2011) and low tidal amplitudes (~20 cm, Figs. 7f and 8b) in this region can place the GL position slightly seaward. Therefore, different locations between ICESat-2-derived Point F and the reference GL (ESA CCI Point F in October 2016 in this region) are possibly caused by different ocean tidal amplitudes. In addition, proving our method works at a 20 cm scale tidal amplitude in a region with complex relief demonstrates the ability for the generation of GLs for the majority of the Antarctic Ice Sheet, from regions with low tidal ranges such as the Amundsen Bay Embayment (< 1 m) to regions with large tidal range such as the Ross Ice Shelf (~1 m – 2 m) and the Filchner-Ronne Ice Shelf (> 4m) (Padman et al., 2002).

On Hearst Island, several Points F were identified from ICESat-2 data. Although no ESA CCI DInSAR Point F is available in this region for comparison, we mapped the distributions of the absolute elevation changes, |dh|, at crossovers (Fig. 9). The transitions from land ice (low |dh|) to floating ice (high |dh|) observed from the crossovers align well with the break-in-slope from the REMA slope map and the distribution of ICESat-2-derived Point F from our repeat track analysis (black dots in Fig. 9). Note the Point F, indicated by the black arrow in Fig. 9, is located slightly landward compared to the REMA break-in-slope. The |dh| of nearby crossovers show a similar pattern, with the transition point between red and blue points locating slightly landward of the REMA break-in-slope. The combination of repeat-track-derived Point F and the distribution of crossover |dh| in this region shows that break-in-slope cannot always effectively represent the actual GL even in high-slope and slow-moving areas. Although the spatial separation between crossovers (Fig. 9) (~3 km in Hearst Island) is far larger than the 20 m along-track separation afforded by repeat track observations, surface elevation changes derived from ICESat-2 crossover data can still provide valuable information about the approximate location of grounding zone. In doing so, this method has the potential to provide important validation of repeat-track-derived GZ features, including fast flowing ice streams where the GZ can undergo rapid changes (Rignot et al., 2011). As one of the northernmost ice shelves in Antarctica, Larsen C Ice Shelf is subject to the highest across-track spacings. Higher density of crossovers will be available further south on the larger ice shelves (e.g. ~ 6 per km$^2$ on the Ross Ice Shelf) as a result of decreased across-track spacing.

The hydrostatic point H mapped from ASAID project by combining ICESat-derived Point H and Landsat-7 imagery is the most complete product for Point H to date (Bindschadler et al., 2011), with a positional error of about 2 km. The absolute mean separation and standard deviation between ICESat-2-derived Point H and ASAID Point H (Figs. 7b and 7d) are 1.2 km and 0.98 km (Table 2), respectively. 83% of ICESat-2-derived Point H locate less than 2 km away from the ASAID Point H (Fig. 7d), which is within the geolocation error of ASAID Point H. The largest deviations occur in Joerg Peninsula (Fig. 7d), which is likely that ASAID Point H is in error here as it fails to capture the concave shape of GZ in this region as shown from the ESA CCI DInSAR Point F in Fig. 7a.

Table 2. Absolute mean separations and standard deviations between ICESat-2-mapped GZ with ESA CCI DInSAR Point F, ASAID Point I and ASAID Point H.

| | | Absolute mean separation (km) | Standard deviation (km) |
|---|---|---|---|
| Point F | DInSAR Point F | 0.39 | 0.32 |
| | ASAID Point I | 0.34 | 0.28 |
| Point H | ASAID Point H | 1.2 | 0.98 |

At the time of this study, only four repeat cycles were available and about a third of the GZ were calculated from two repeat cycles (Fig. 7e). Changes in ocean tide amplitude can induce short-term changes of grounding line position up to 4 km at different tidal levels (Milillo et al., 2017), depending on the acquisition time of the repeat cycles. Although this short-term tidally induced GL migration is not significant over the long observation period with large GL retreat (Rignot et al., 2014), it may hide the real GL retreat signal on a sub-annual scale (Milillo et al., 2017). The capability of our method of identifying the GZ using only two repeat cycles can be used to detect the short-term GL changes caused by ocean tides, and allows us to distinguish the actual GL retreat signal from this tidally induced GL migration with more ICESat-2 repeat cycles available in future.

**4.3 Grounding zone width**

The width of the GZ depends on ice stiffness, bed slope and ice thickness (Bindschadler et al., 2011), and is useful in determining the ice thickness and rheology across the grounding zone (Dawson and Bamber, 2020). Since the orientation of ICESat-2 ground tracks are not always perpendicular to the GL, calculating the along-track distance between Points F and H as the GZ width can overestimate the actual value. Here we adopted a similar method used in Brunt et al. (2010b) by converting the along-track distance to the cross-GL distance. By assuming that the reference GL generated in section 3.1.2 provides a good reference for the local orientation of the actual GL, we calculated the GZ width at each Point F to be the length of the perpendicular line from Point H to the tangent line of the reference GL at the intersection between the nominal reference track and this reference GL. The width of 221 GZs were calculated in this study (Fig. 10), which varies from 0.31 km to 11.62 km with an average of 2.51 km, and the standard deviation is 1.61 km. The distribution of GZ width agrees well with the CryoSat-2-derived GZ width shown in the Fig. 6d of Dawson and Bamber (2020). Similarly to the GZ width distribution on Filchner-Ronne Ice shelf from ICESat (Brunt et al., 2011), there is only one GZ width exceeding 10 km. This GZ locates inside a concave inlet of a glacier in Churchill Peninsula (Fig. 10). The large GZ width could be related to the ice thickness and geometry of GL in this location, but a detailed investigation of the underlying reasons exceeds the research scope of this study.

# 5 Conclusion

We have presented a new, fully automated method of mapping GZ features including Points F and H from ICESat-2 repeat track and crossover data. This method addresses the issue of residuals in elevation change caused by large across-track slopes and the results are partially validated by a crossover analysis of ascending and descending tracks. Using a 10-month period of ICESat-2 ATL06 data spanning from March 2019 to March 2020, we are able to map the majority of the grounding zone in Larsen C Ice Shelf, including highly crevassed regions such as Jason Peninsula and Churchill Peninsula, and the rarely investigated regions such as Bawden and Gipps Ice Rise, which are the important pinning points for Larsen C Ice Shelf. 253 Point F and 263 Point H were identified in Larsen C Ice Shelf, representing a near nine-fold increase compared with the GZ features mapped from ICESat. Our ICESat-2-dervied GZ features agree well with previous measurements. The mean absolute separation and the standard deviation between ICESat-2-derived Point F and ESA CCI DInSAR Point F are 0.39 km and 0.32 km, respectively. The mean absolute separation and the standard deviation between ICESat-2-derived Point H and ASAID Point H are 1.2 km and 0.98 km, which are within the ~2 km positional error of ASAID Point H. The lowest tidal range, detected by our method on Larsen C Ice Shelf is ~20 cm, making it possible to apply the repeat track analysis to other regions with low tidal range, such as the Amundsen Sea Embayment. In addition, the ability of mapping the GZ using only two repeat cycles should allow the detection of short-term GL changes caused by ocean tides and the separation between the long-term GL retreat signal from the short-term tidally induced GL migration when more repeat cycles are available in future. Although the distribution of elevation change from a crossover analysis depends on the across-track spacing of ICESat-2 ground tracks, the example of Hearst Island indicates that the crossover analysis can show the approximate location of the GZ. With smaller across-track spacings in southern regions of the Antarctic Ice Sheet, similar crossover-based analyses have the potential to provide more accurate depictions of the GZ.

*Data availability.* The ICESat-2 data and ICESat grounding line data used in this study are available from the National Snow and Ice Data Center (NSIDC). The datasets generated in this study are available from the authors upon request.

*Author contribution.* TL developed the methods and wrote the paper. GJD and SJC assisted with data processing. JLB conceived the study. GJD and JLB contributed to the interpretation of the results. All authors commented on the manuscript.

*Competing interests.* The authors declare no completing interest.

*Acknowledgments.* The authors would like to thank the two anonymous reviewers for their valuable comments and suggestions, which greatly improved this paper. We thank the European Space Agency (ESA) for providing the ESA CCI DInSAR grounding line data. This work was supported by the China Scholarship Council (CSC) - University of Bristol 380 joint-funded PhD Scholarship.

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

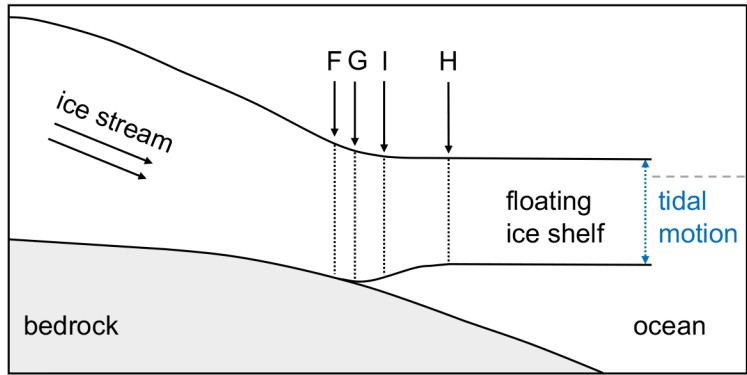

**Figure 1. Schematic of the ice shelf grounding zone structure adapted from Fricker and Padman (2006). Point G is the true grounding line, Point F is the landward limit of ice flexure induced by tidal motion, Point H is the seaward limit of ice flexure and the inshore limit of hydrostatic equilibrium, Point I is the break in surface slope.**

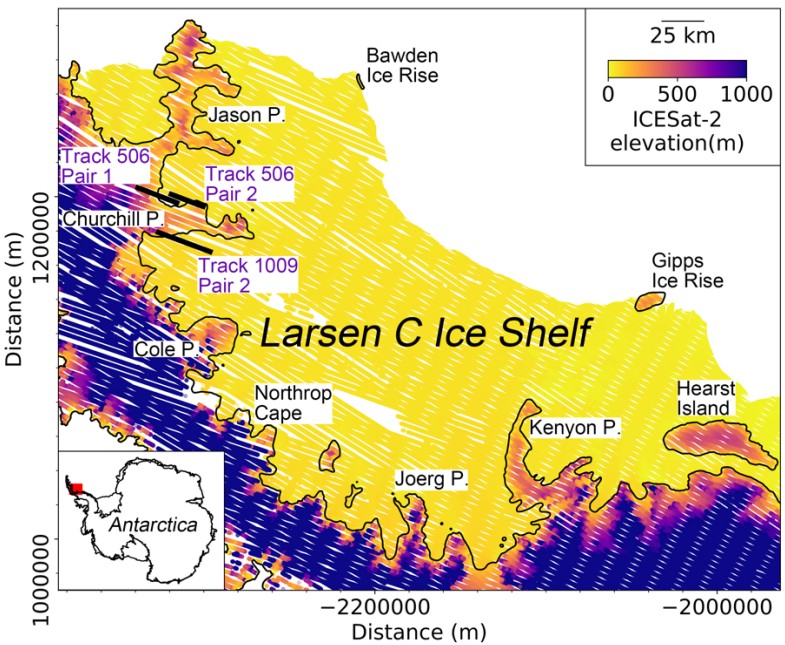

**Figure 2. Locations and surface elevations of ICESat-2 ground tracks from repeat cycles 3, 4, 5 and 6 on Larsen C Ice Shelf in**
**Antarctic Polar Stereographic (epsg:3031) projection. The black line is the grounding line (GL) from Depoorter et al. (2013). Red box on the inset map shows the location of Larsen C Ice Shelf.**

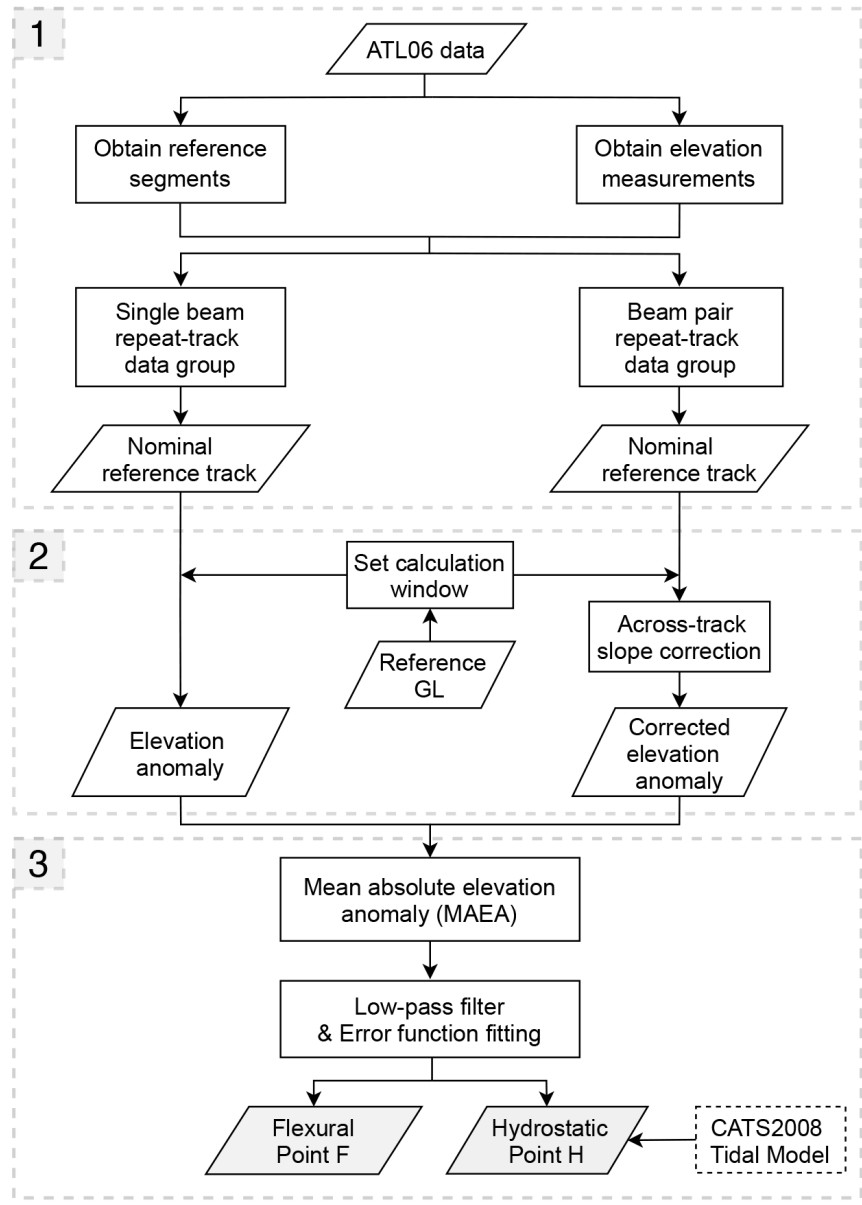

**Figure 3. The ICESat-2 repeat-track workflow used to identify the grounding zone (GZ) in this study, including the limits of inland tidal flexure (Point F) and hydrostatic equilibrium (Point H).**

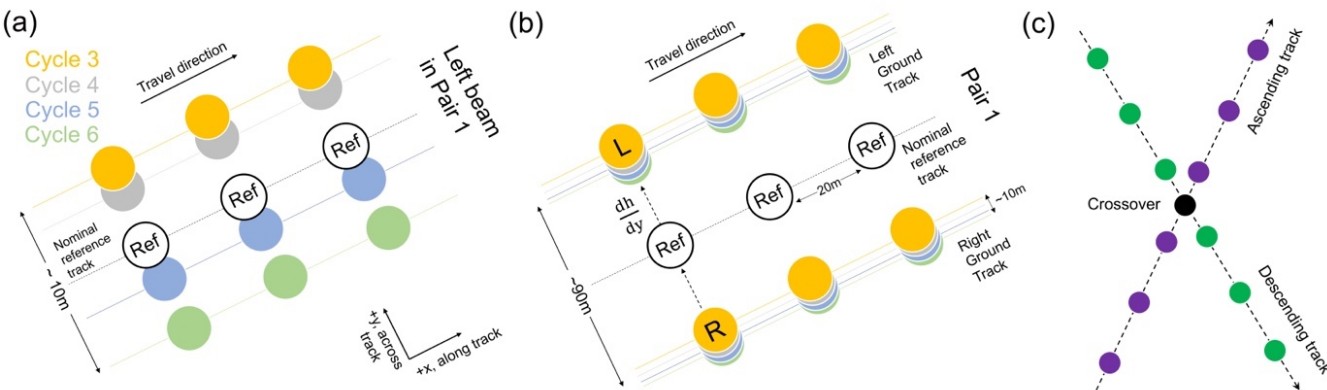


Figure 4. Schematics for repeat track analysis and crossover analysis. a) Repeat track analysis method for left beams in beam pair 1 (out of 3 beam pairs) from four repeat cycles 3, 4, 5 and 6. Black circles are the nominal reference track. b) Repeat track analysis method for both left and right beams in beam pair 1 from four repeat cycles. Within each repeat cycle, elevations of two beams in a pair were corrected for across-track slope onto the nominal reference track in the middle. c) Crossover method showing the
interpolation along the ascending track (purple dots) and the descending track (green dots) to their intersection as the crossover (black dot).

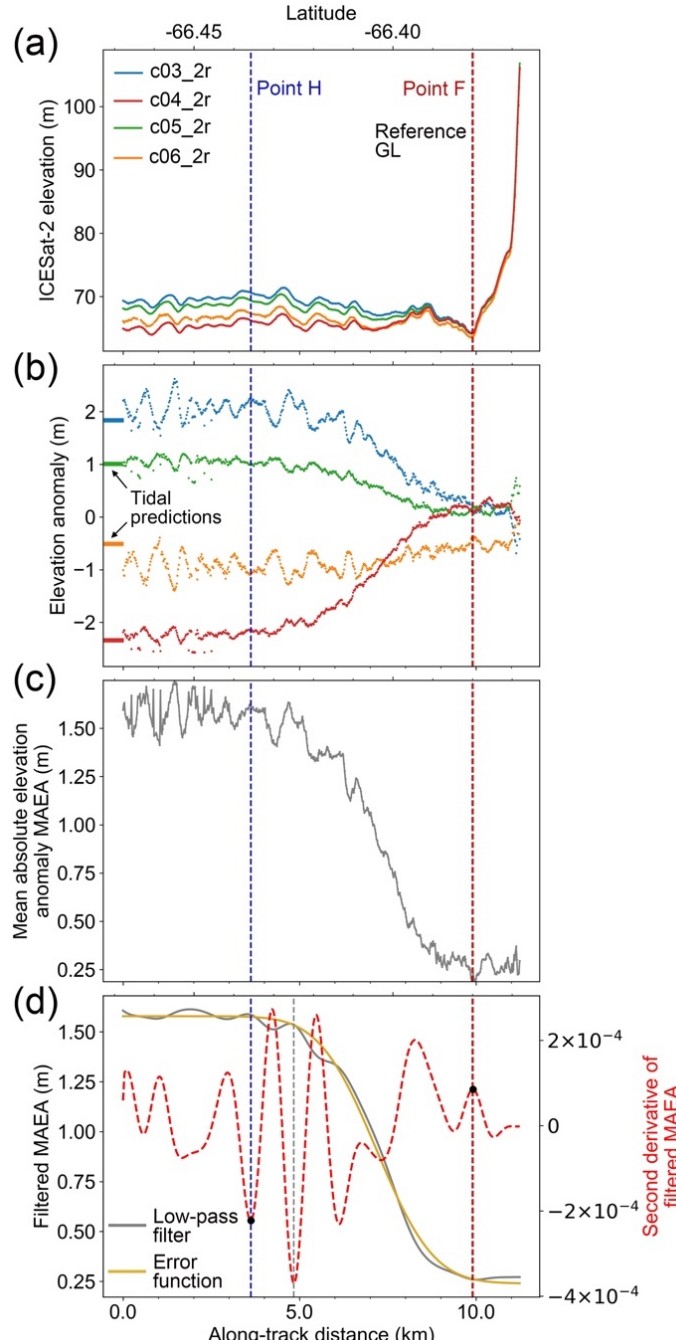

**Figure 5. ICESat-2 repeat track analysis for four right beams from repeat cycles 3, 4, 5 and 6 in beam pair 2 of track 1009. The location of track 1009 beam pair 2 is shown in Figure 2. a) ICESat-2 're-tided' elevation profiles. c03_2r in the legend refers to the right beam of beam pair 2 in repeat cycle 3. b) The elevation anomalies of each repeat track. Horizontal lines at the left are the zero mean tide height predictions from the CATS2008 Tidal Model (Padman et al., 2002) following Fricker and Padman (2006). c) The Mean Absolute Elevation Anomaly (MAEA). d) Low-pass filter filtered MAEA is shown in grey solid line, Error function**


fitting of the MAEA is shown in yellow solid line, the second derivate of low-pass filter filtered MAEA is shown in red dashed curve, the black dots are the locations of inland limit of tidal flexure Point F (right) and inland limit of hydrostatic equilibrium Point H (left), the vertical dashed grey line is the location of Point H when using the third derivate of Error function as the guide point. Locations of Point F, Point H and reference grounding line (GL) are marked as the vertical dashed red line, vertical dashed blue line and vertical dashed black line in all subplots.

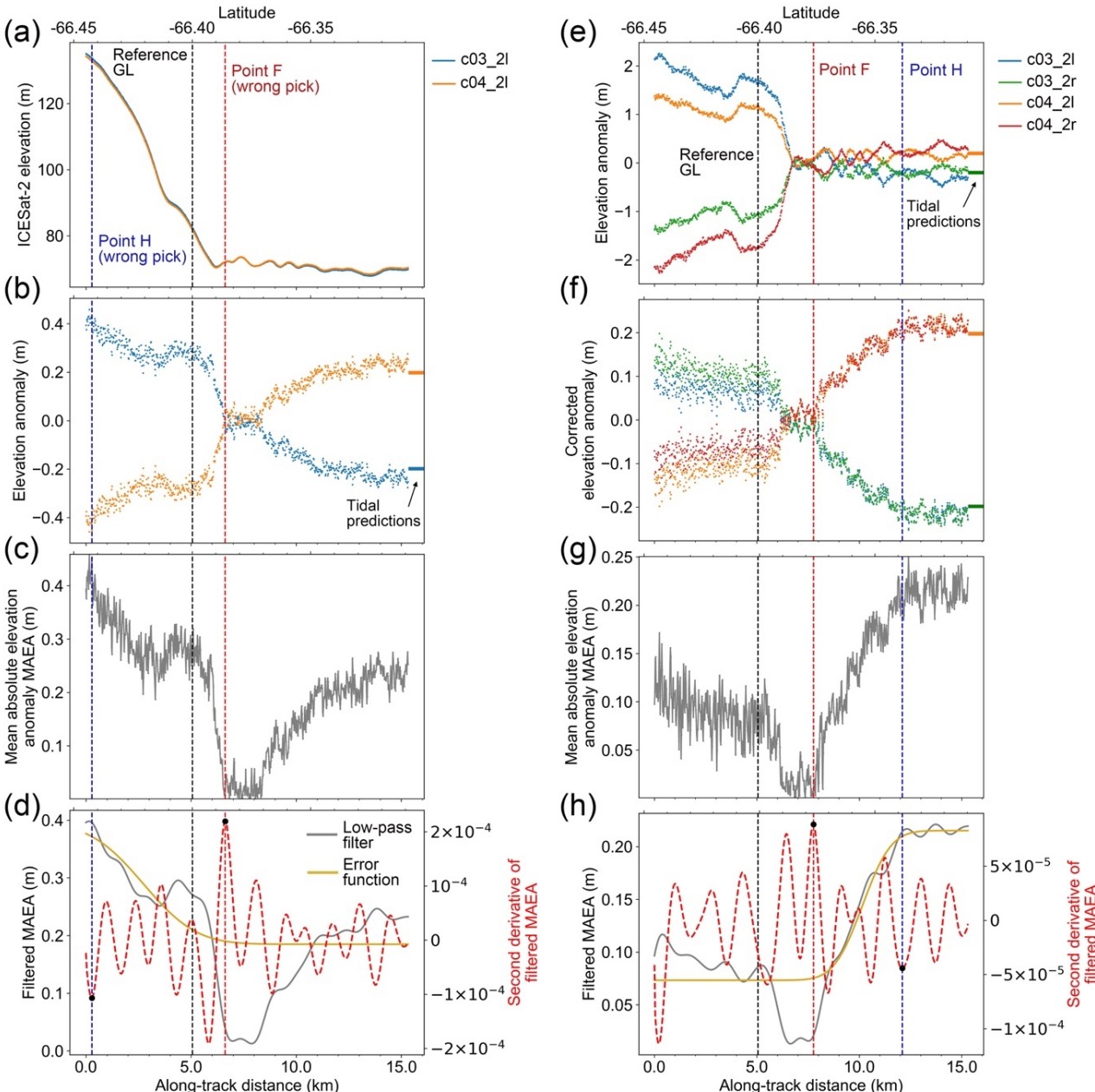

Figure 6. Comparison between repeat track analysis for single beams and beam pairs. a-d) ICESat-2 repeat track analysis for two left beams from repeat cycles 3 and 4 in beam pair 2 of track 506. The location of track 506 beam pair 2 is shown in Figure 2. Same as Figure 5, but showing the wrong grounding zone (GZ) features picked by repeat track analysis due to large across-track

slope on land ice. e-f) ICESat-2 repeat track analysis for both left and right beams in beam pair 2 of track 506 from repeat cycles 3 and 4. e) The elevation anomalies of each track. f) The corrected elevation anomalies after across-track slope correction. g) and h) same as Figure 5c and 5d.

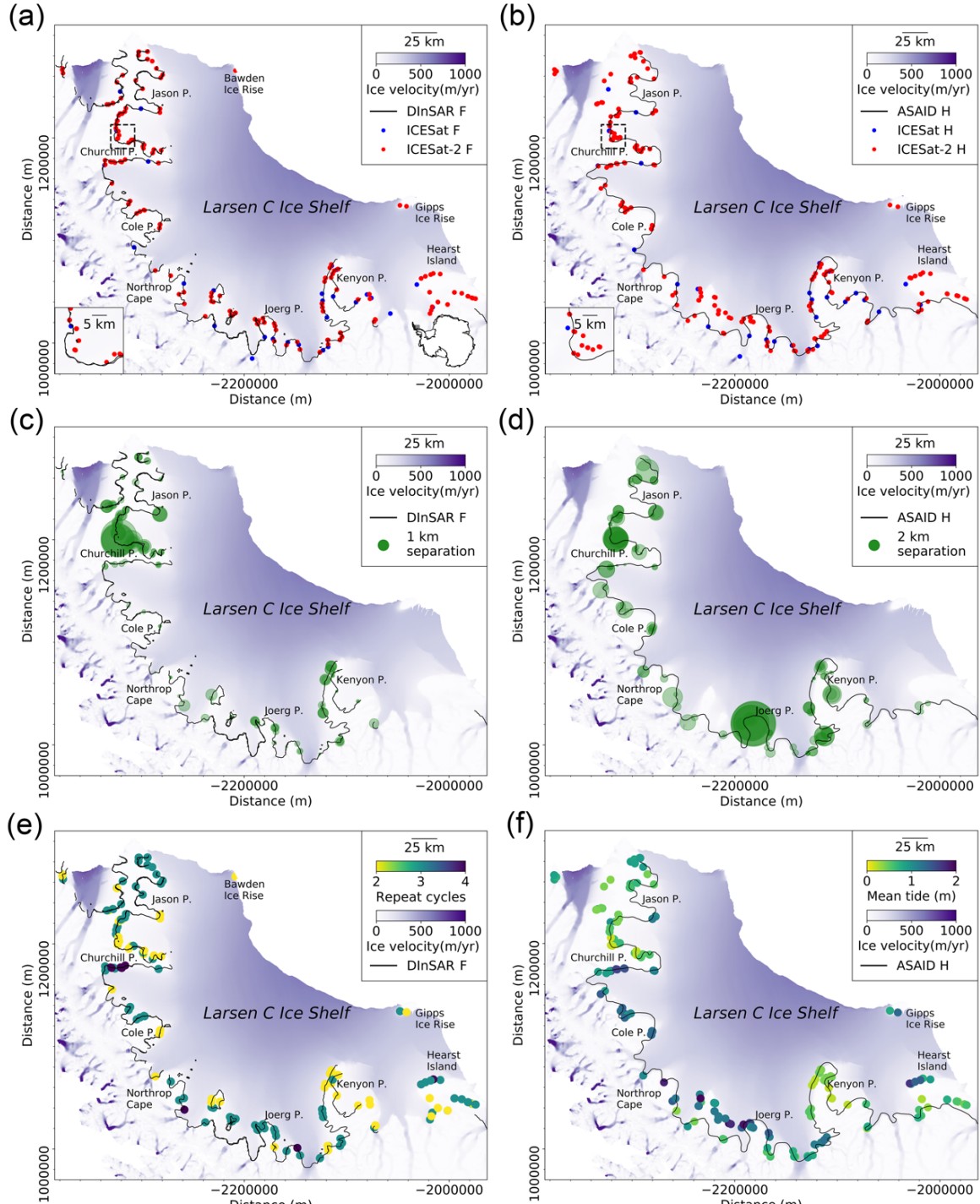

**Figure 7. Spatial distributions of ICESat-2-derived GZ features. a) ICESat-2-derived inland limits of tidal flexure (Point F; red dots). For comparison, ICESat-derived Point F (Brunt et al., 2010a) are also shown (blue dots), ESA CCI DInSAR-derived Point F is shown as the black line (ESA Antarctic Ice Sheets CCI, 2017). b) same as a), but showing ICESat-2-derived inland limit of hydrostatic equilibrium (Point H; red dots) and ICESat-derived Point H (Brunt et al., 2010a) (blue dots), Point H from the ASAID grounding line project is shown as the black line (Bindschadler et al., 2011). c) Absolute separations between ICESat-2-derived Point F and ESA CCI DInSAR-derived Point F. d) Absolute separations between ICESat-2-derived Point H and ASAID Point H. e) Number of repeat cycles used to calculate the grounding zone (GZ) features. f) Distribution of mean ocean tide range at Point H. In all subplots, data are superimposed over recent ice surface velocity magnitudes (Rignot et al., 2017) in Antarctic Polar Stereographic (epsg:3031) projection.**

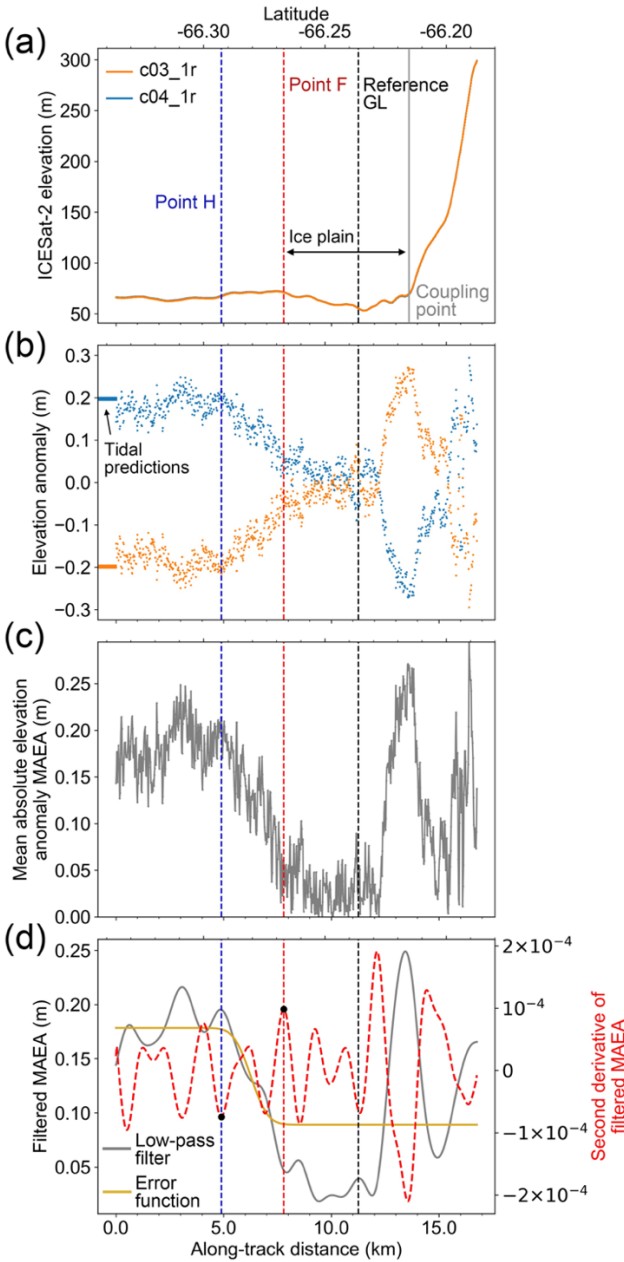

**Figure 8. ICESat-2 repeat track analysis for two right beams from repeat cycles 3 and 4 in beam pair 1 of track 506. The location of track 506 beam pair 1 is shown in Figure 2. Same as Figure 5, but showing the existence of an ice plain between the inland limit of tidal flexure (Point F) and the coupling point.**

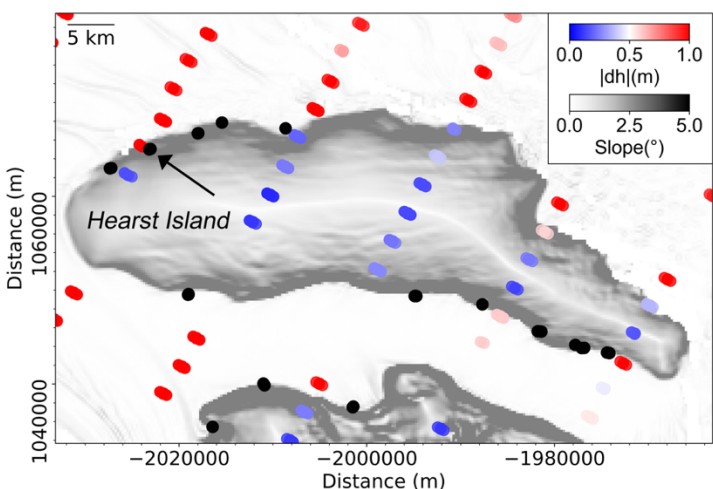


**Figure 9. Spatial distribution of ICESat-2 crossovers in Hearst Island. The absolute change in elevation at each crossover, |dh|, is shown as the color-coded dots. ICESat-2-derived inland limit of tidal flexure (Point F) is shown as black dots. Background is the surface slope from the REMA DEM with a 250 m resolution (Howat et al., 2019) in Antarctic Polar Stereographic (epsg:3031) projection.**

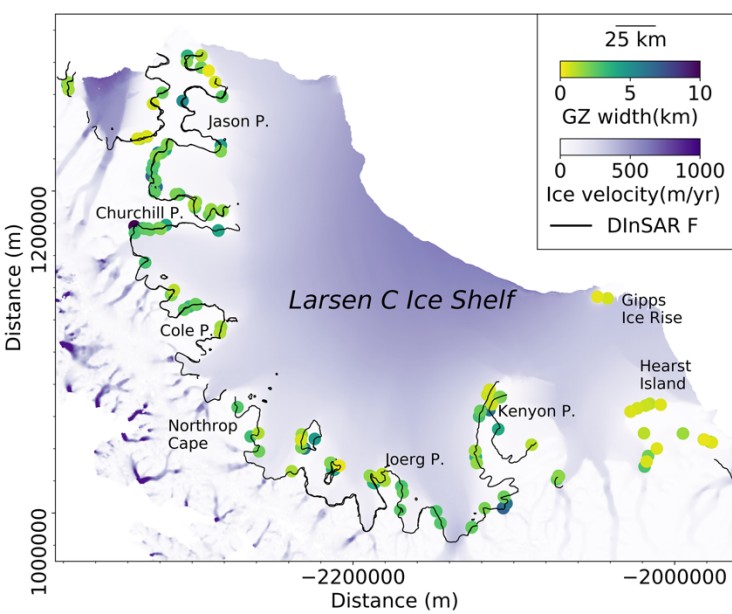


**Figure 10. Distributions of GZ width. ESA CCI DInSAR-derived inland limit of tidal flexure (Point F) is shown as the black line (ESA Antarctic Ice Sheets CCI, 2017). Background is the recent ice surface velocity magnitudes (Rignot et al., 2017) in Antarctic Polar Stereographic (epsg:3031) projection.**

# Appendix A

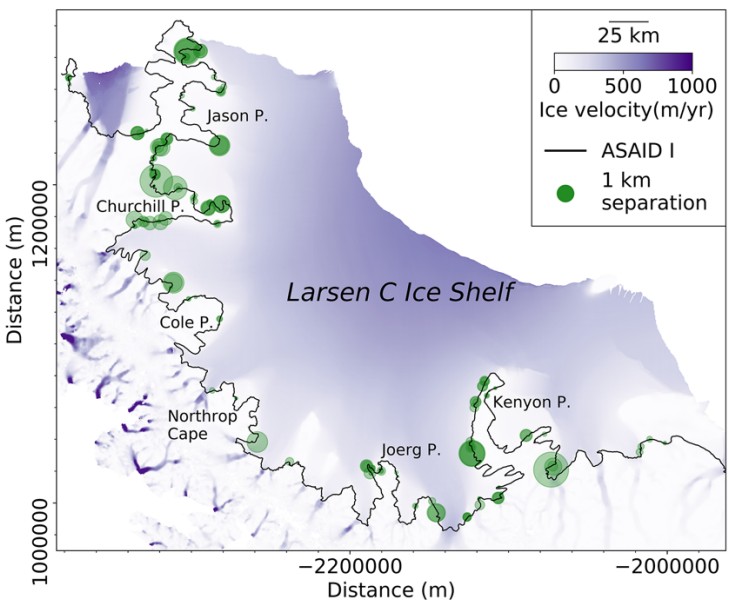

**Figure A1. Absolute separations between ICESat-2-derived inland limit of tidal flexure (Point F) and ASAID break-in-slope point (Point I). ASAID Point I is shown as the black line (Bindschadler et al., 2011). Background is the recent ice surface velocity magnitudes (Rignot et al., 2017) in Antarctic Polar Stereographic (epsg:3031) projection.**

