# Peer review of "Mapping the Grounding Zone of Larsen C Ice Shelf, Antarctica, from ICESat-2 Laser Altimetry"

_The Cryosphere, 2020_

## Referee Comment (RC1) · Anonymous Referee #1 · 15 Jun 2020

**Review of "*Mapping the Antarctic Grounding Zone from ICESat-2 Laser Altimetry*" by Tian Li et al.**

Li et al. use 12 months of observations from the newly launched ICESat-2 laser altimeter to recover two key components of Larsen C Ice Shelf's grounding zone (GZ). Building upon the repeat-track methodology first applied to ICESat GLAS data (e.g. Fricker & Padman, 2006; Fricker et al., 2009; Brunt et al., 2010; 2011), the authors use elevation departures from mean tidal state to infer the landward limit of tidal flexure and point of hydrostatic equilibrium over successive repeat passes. By exploiting the beam-pair imaging capabilities of the ATLAS instrument on-board ICESat-2, the authors also apply a cross-track slope correction to their dataset, greatly improving the signal-to-noise ratio of GZ detection over areas of rough terrain compared with single-beam ICESat methods. Elevation changes derived from ICESat-2 ascending and descending cross-over passes are also used to approximate the position of the GZ (although at much degraded resolution c/w their repeat-track observations), thereby acting as a part-validation of their repeat-track-derived GZ picks.

Given the close agreement between ICESat-2 and independent InSAR-based GZ observations (~+/- 0.5 km separation, assuming no GL migration between acquisition dates) and the fact that this technique is fully automated, I believe that Li et al's technique will be of large interest to the readership of *The Cryosphere*, and look forward to seeing this paper published. The prospect of applying this technique to the entirety of Antarctica's grounding zone is also an exciting one. Before acceptance, however, I feel that moderate revisions must be implemented to improve the clarity and flow of the manuscript, and to address several technical points pertaining to the GZ detection technique and comparison with other data sources. My suggested revisions and other comments are included below.

**General / Major Comments**

While it's clear that the authors have invested a large amount of time and effort in the development of their GZ detection technique, I feel that large parts of the manuscript should be thoroughly edited by all authors to increase the overall clarity, flow and standard of English presented in the text. At present, the main text is at times highly verbose and/or repetitive, which – together with the use of non-standard sentence/paragraph structure throughout the manuscript – makes it difficult for the reader to easily follow the narrative of the paper. This is especially true in the methodology and results sections, where, for example, the reader must repeatedly flick back and forth between the discussion of Points F and H - this has the potential to be extremely confusing, especially to those unfamiliar with the anatomy of the GZ. As I outline in my specific comments, these sections could easily be restructured to offer a more streamlined discussion. I would also strongly encourage the authors to correct the numerous instances of incorrect grammar contained in their manuscript (I have indicated some, but not all instances of this in my specific comments).

***Lines 183-216*** – To my knowledge, the 2017 ESA CCI dataset provides near-complete coverage of Larsen C's GL as constrained from Sentinel-1a/b DInSAR-based observations acquired in 2016/2017. I am therefore surprised to read that the majority of the ICESat-2-DInSAR comparisons made in this manuscript refer to the earlier (mostly 1990s ERS-1/2-based) GL products provided by Rignot et al., (2016), and not those more temporally contemporaneous with the authors' ICESat-2 observations.

Furthermore, while the authors correctly mention in lines 106/107 that recent (2010-2016) GL migration has been fairly minimal, a comparison of the Rignot and ESA CCI products reveal large retreats (up to 4 km) along parts of Larsen C's GL (including Churchill and Kenyon

Peninsulas), presumably in response to the sustained period of ice-thinning between 1979 and 2008 noted in the literature (Fricker et al., 2012, JGR: Oceans;  Adusumilli et al., 2018, GRL). (NB. Since 2008, Larsen C has largely thickened, consistent with the more stable rates of GL migration reported by Konrad et al. (2018)).

As such, I believe making primary reference to the old Rignot GL dataset is inappropriate in this context, and would strongly suggest the modification of the abstract, lines 183-216, discussion/conclusion and Figures 2, 5, 6 and 7 to refer only to the newer ESA CCI GL.  Also, be sure to include the appropriate citation for the ESA CCI dataset in the reference list.

**Technical Comments**

**Line 71 and Figs 5 and 7** – The authors mention the use of data from repeat cycles 3-5 in their repeat track analyses, but Figures 5 and 7 only show the solution as derived from repeat tracks 3 and 4. Why is this? For consistency with the main text, I'd update Figures 5 and 7 to show the full complement of data acquired over the entire range of images tidal states.

**Line 102 –** As the authors know, the Depoorter GL is a compilation of multiple GL datasets, derived from a variety of techniques (e.g. InSAR, break-in-slope, ICESat) applied to satellite imagery acquired at different times. For absolute clarity here, I'd consider adding an overview of the imaging times and techniques used over this region of Antarctica.

**Lines 120-125 –** For clarity, I suggest defining each of the terms included in Eq. 1 before discussing Eq. 2. Also note the lack of any explicit definition of $Y_{ATC,G}$ in the current version of the manuscript.

**Lines 136-160 –** In Figure 1 the authors show the position of $I_b$, the ice-sheet-shelf inflexion point (otherwise known as the break-in-slope), which is obtainable from optical-, DEM- and altimetry-based methods. As the authors are aware, $I_b$ has been used extensively in the literature (including Christie et al. (2016) as mentioned in the text, but also e.g. Bindschadler et al. (2011, *The Cryosphere*) and Scambos (2007, *Remote Sensing of Environment*)) to supplement our understanding of the GZ in areas where DInSAR coverage is poor. I was therefore a little surprised to see no discussion or method of deriving this important GZ component using ICESat-2 in the text. This is despite the break-in-slope being clearly visible in e.g. Figures 5a and 5c. Was any attempt to automate this component of the GZ made? If not, why not? I think a brief discussion of $I_b$, its manifestation in the ICESat-2 data and any attempts to automate it should therefore be added to the Section 3.1. If easily obtainable, then an additional figure similar to Figure 5c could also be added to show any differences in ICESat-2-derived $I_b$ and the break-in-slope mapped by ICESat-1 (Brunt et al., 2011) and/or the ASAID team (Bindschadler et al. 2011).

On a related note, I have noticed occasional large errors in the positioning of DInSAR-derived GLs (Rignot, ESA CCI) over the Antarctic Peninsula (including Larsen C), presumably related to the geo-coding of SAR data over the Peninsula's steep terrain using old DEMs, and/or the misattribution of the Point *F* on ice landward of nunataks or other mountainous exposures at the GZ. These positional errors are not seen in e.g. the ASAID product due to the on-nadir viewing angle of Landsat and the ability to distinguish between mountains and ice more easily. Therefore, any comparison of ICESat-2- and ASAID-derived $I_b$ picks may be more representative of the true GL (in lieu of Point *F*) in these areas. (As a relatively slow flowing

region with a well-defined break-in-slope, $I_b$ will fall close to Point $G$ (as it does in Fig. 5), making it reliable proxy for Point $G$ and $F$).

In light of the above and if not already done, the authors should thoroughly QC the DInSAR product for such offsets – their omission could actually improve your mean absolute separation statistics!

***Lines 150 and 156*** – How sensitive is the pinpointing of Point $H$ to the prescribed variance threshold used in the error function, and why was the fourth derivative of this function chosen? In other words, how would e.g. a 3$^{rd}$ order fit look on Figure 5d, and where would $H$ be placed as a result? Some discussion of your choice is needed in the text here.

***Line 190*** – With the exception Churchill Peninsula (Figure 6a) where *****one***** ICESat-1 $F$ point falls very close to the ICESat-2 $F$, spatially coincident ICESat-ICESat-2 observations do not exist anywhere else along the Larsen C coastline. I would therefore excise any comparison between the two datasets from the text as I don't think such statements can be made with confidence. Instead, I'd solely report on the close agreement between ICESat-2 and DInSAR observations. (the discussion of the enhanced coverage between ICESat-1 and ICESat-2 is okay, however!).

***Sentence beginning line 196*** – The GZ information contained in Dawson and Bamber (2017) refers to Siple Coast, not Larsen C. Are the reported separation values mentioned in the text derived from those observations over the Siple coastline, or has the methodology of Dawson and Bamber been extended to Larsen C? This is essential information to state here.

***Lines 217-226*** – In Figure 8 (b and c especially), I see numerous instances of dark blues over the ice shelf which, following the main text, readers might incorrectly interpret as grounded ice. The opposite is true over grounded ice, which may be interpreted to be floating. What is the explanation for these observations? Surface processes? A few sentences explaining these blips are essential here.

***Lines 233-235*** – Similar to my comments on the potential derivation of $I_b$ from ICESat-2 data and its comparison with e.g. the ASAID product, I am surprised to see no comparison of the authors results with ASAID-derived $H$. At present this our best pan-ice sheet understanding of $H$, so it would be very useful to see a discussion of how your results compare here.

***Line 275*** – As no reference is included to support this assertion, the discussion of crevasses and their influence of GZ reads like something of an afterthought, potentially degrading the reader's confidence in this otherwise highly impressive technique. Since this is predominantly a methods paper, I'd encourage the authors to elaborate on a) why they didn't (couldn't?) clean up these data, b) exactly how they think these phenomena may influence the overall accuracy of their technique and, if needed, c) briefly mention how future work may address this issue. I imagine a lot of this could (and should) go into Section 3 so that the reader can bear these uncertainties in mind while reading Section 4. In addition, it is generally poor grammatical form to begin new sentences with 'Also', 'However', 'It' etc., so the readers should consider revising the structure of the next sentence (and several other similar instances throughout the manuscript) to address these minor niggles.

***Line 277*** – Good point. Does the processing chain provide a means in which the user can obtain the number of repeat-track observations your GL picks are based on? Such information could be used to calculate a 'GZ pick reliability rating', and/or inform data users about the range of tidal frequencies the GZ is mapped over.

**Specific Comments**

**Line 6 –** While it's beyond the scope of this review to provide detailed corrections to the style of English and grammar used throughout the manuscript, I've re-structured the abstract to greatly condense and more effectively convey the key selling points of paper here. The authors should feel free to amend/disregard as necessary. Authors should also note the omission of greater 'precision' relative to ICESat-1, as I strongly believe this is not convincingly addressed in the manuscript (see my comment above on line 190).

"*We present a new, fully automated method of mapping the Antarctic Ice Sheet's grounding zone using repeat-pass and cross-over analyses of newly acquired ICESat-2 laser altimeter data. Our method recovers the position of the landward limit of tidal flexure and the inshore limit of hydrostatic equilibrium, as demonstrated over the mountainous and hitherto difficult to survey grounding zone of Larsen C Ice Shelf. Since launching in 2018, our method has already doubled the number of grounding zone observations acquired from ICESat-1, which operated between 2003 and 2008. Acting as a reliable proxy for the grounding line, which cannot be directly imaged by satellites, our ICESat-2-derived limit of tidal flexure locations agree well with independently constrained measurements, with a mean absolute separation and the standard deviation of 0.29 km and 0.31 km, respectively, from interferometric synthetic aperture radar-based observations. Our results demonstrate the efficiency, high spatial precision and density in which ICESat-2 can image complex grounding zones, and its clear potential for future pan-ice sheet grounding-zone mapping efforts.*"

**Lines 18-21 –** I think these opening sentences could be worded more eloquently to say something like: "*Long-term satellite observations have linked the on-going thinning of Antarctica's ice shelves (Paolo et al., 2015) to enhanced rates of ice mass discharge beyond the grounding line (hereafter GL) – the point where the grounded ice sheet first detaches from the bedrock and begins to float (Fricker & Padman, 2006). Ice discharge calculations are sensitive …*".

**Line 28 –** Provide references for mass balance studies and those assessing overall ice sheet stability. The latest offering from the IMBIE crowd (Shepherd et al., *Nature*, 2018) would be good here.

**Line 33 –** Following my comment regarding Line 275, the authors should aim to avoid ambiguous terms like '*Its*' to begin sentences. Suggest rewording to "*The precise width of the grounding zone depends on …*".

**Lines 35-36 –** This is a good example of many instances of repetition in the manuscript. The true GL (*G*) and limits of tidal flexure (*F*) have already been defined on line 30/31, so don't need to be repeated here. Saying something like the following would be much more concise and easier to understand.

"*… While Point G cannot be detected directly from satellite-based observations, Point F resides within close proximity of this location, and is thus widely considered to be the most robust satellite-observable proxy for the "true" grounding line (insert references here).*"

(Note that this rewrite also gets round the incorrect notion that the limit of tidal flexure is something only present on the ice sheet surface, when reality it is the surface manifestation of a variety of mechanical processes going on at the ice-bed-ocean interface)

I also suggest you replace any further mention of "*the true grounding line*", "*the landward limit*", "*limit of hydrostatic equilibrium*" etc. in the manuscript with 'Point *G*', 'Point *F*' and 'Point *H*'.

Also, in this instance '*Point*' is a pronoun, and so should always be capitalized. (be sure to change this universally throughout the text and fig. captions).

**Line 38 –** Remove '*to date*' (as it probably always will be …).

**Line 41 –** Change to "*Points F and H (Figure 1) have previously also been derived using ICESat-1 laser altimetry …*"

**Line 43 –** The use of the word '*these*' is unneeded and can be removed.

**Line 44 –** Suggest replacing '*data*' with "*additional GZ information across regions …*". Also hyphenate '*DInSAR-derived*'. For consistency with the above paragraph, please also refer to '*DInSAR-derived GZ information*' rather than '*GL information*' (since DInSAR recovers *F*, a \*proxy\* for *G*).

**Line 46 –** Change to "*… In addition, ICESat-1-based GZ detection is heavily based on visual interpretation which requires a large amount of manual …*" for brevity.

**Line 51 –** Poor grammar. Suggest rewording and merging with the next sentence for clarity and conciseness. Something like: "*ICESat-2 has a repeat-pass orbit time of 91 days, and – compared to the single Geoscience Laser Altimeter System (GLAS) onboard ICESat-1 – measures the surface elevation of Earth's ice sheets using three photon-counting beam pairs emitted from the Advanced Topographic Laser Altimeter System (ATLAS)*".

**Line 58 –** Suggest shortening to: "*… Point H by analyzing repeat-track and ascending/descending cross-over data. We chose the …*". The following sentence is also very long and verbose, and should be condensed and/or split up for clarity.

**Line 64 –** Saying that you used 12 months of data (5 cycles) when in reality the bulk of your analyses only used 8 (3 cycles) is somewhat misleading, so I'd consider splitting this paragraph up into 2 parts. One part discussing the repeat track analyses (8 months of data), and a standalone, follow-up part discussing the cross-over analysis (extended dataset covering 12 months). At present there's a lot of back-and-forth discussion between techniques, which took me several readings to fully understand what you meant.

**Line 69 –** For absolute clarity, do these RPT's refer simply to the 'Pair Track' files included in the ATL06 products, or is this something you have calculated the position of? If the former, then I'd suggest just referring to them as 'pair tracks' (PTs) for consistency with ATL06 naming convention, and to avoid any possible confusion with the Reference Ground Track (RGT) information provided as part of the ICESat-2 orbit files (see: https://icesat-2.gsfc.nasa.gov/science/specs).

**Line 83 –** Should this say +/-15 m as opposed to 150 m? This value seems excessive, and I wonder how many values actually fell out with this range after your initial cleaning of the data?

**Lines 87-90 –** Check grammar, punctuation and sentence tense. The first sentence is also repetitive and could easily be reworded to avoid double use of '*repeat track*'. Also, how/why does your cross-track slope correction 'facilitate' the automation of GZ mapping? Do you mean that it reduced the occurrence of incorrectly identified GZ features? I think a little more elaboration is needed here to explain this point. Sentence beginning line 90 should also be reworded to avoid the use of the word '*It*'.

**Line 91 –** Incorrect use of semi-colon and multiple unneeded uses of the work '*the*'. Also suggest changing '*features estimation*' to '*feature identification*'.

***Paragraph beginning Line 93 –*** This is a good example of the 'back-and-forth' organization of the paper I mentioned in my general comments. That is, whereas reader expects this paragraph to involve a discussion of steps 1 and 2 detailed in line 91, the reader now has to go back to a discussion of repeat cycles and the grounding line reference product. In accordance with processing steps shown in Figure 3, I suggest the authors separate this part of the methods section into several distinct subheadings possibly labelled "*3.1.1: Repeat-track data preprocessing*" (to include discussion of nominal ref track calculation and search window size from reference GL dataset), "*3.1.2: Cross-track slope correction*" and "*3.1.3: GZ component identification*". If the authors choose to do this, then each section should also be explicitly cross-referenced in line 91. The processing steps corresponding to each sub-section could also be indicated on Fig. 3 by enclosing each sub-section in a box.

***Lines 93-96 –*** Be sure to emphasize here that you used all three beam pairs in your analyses, as readers may interpret from the current version of the text that only one beam was used.

***Line 99 –*** As per my comment on line 69. Should '*RPT*' be '*PT*'? Also consider putting inverted commas around '*nominal reference track'* to emphasize that this is a newly introduced term.

***Line 101 –*** Needed to do what? I suspect set a GZ detection search window size, but this isn't obvious from the text.

***Line 103 –*** Why was the Mouginot dataset used over the ice rise information documented in Depoorter? Is it more accurate? More up-to-date? Explicitly state why you used this product here.

***Line 104 –*** What is an '*initial GL product'*? I see no mention of it in the text or Figure 3, and don't know how it differs from the 'reference GL product' discussed in the same sentence.

***Line 111 –*** In 2017 Larsen C calved massive iceberg A-68 from its ice front, exposing an ~6000 km2 area of previously ice-shelf covered ocean water. Mouginot's coastline precedes this calving (and the ICESat-2 observational period), meaning that the authors have inadvertently included this huge section of open water in their analyses. This is clearly seen in Figure 2. I suggest the authors re-clip their dataset using a more up-to-date coastline, such as the one available at: https://www.add.scar.org/. The authors may also wish to consider clipping the velocity dataset shown in Figure 5 for consistency between figures.

***Line 112 –*** This sentence is highly repetitive and includes '*elevation change'* three times. Consider condensing.

***Line 119 –*** Insert comma after $h_{Li,G}$. Also, I presume here that '*Li*' means linearly interpolated? Your choice of interpolation isn't explicitly stated here, so if so, include that info here.

***Line 125 –*** "… *are the y coordinates measured perpendicular to the RGT*". Can this be reworded to say "… *are the across-track y-coordinates measured perpendicular to the RGT*"? (I guess, after all, that this is what '*ATC*' in e.g. '*$y_{ATC, x}$*' means?)

Also, following my comments on Line 69, the use of *RGT* here particularly qualifies the need to abbreviate '*Reference Pair Track (RPT)*' to '*PT*' in the text.

***Line 127 –*** Suggest changing *"in GZ estimation"* to *"used in our identification of the grounding zone".* Remove unneeded period at the end of the last sentence.

***Line 129 –*** Suggest changing to *"After applying the cross-slope track…".* Also consider splitting this very long sentence in two between '*track*' and '*which*', and opening the new sentence with "*These profiles are shown in Figure 5a*".

Also note here the irregular ordering of the Figures in text. Figures should generally be referenced in the text in alphabetical order, so in this instance I suggest swapping Figures 5a and b in Figure 5, and changing Line 131 to say '*5a*'. Same for line 132 ('*5b*'). Also, remove "*color-coded dot curves*" here and in line 132; this is non-essential information best left for the caption.

**Line 133 –** Suggest: "*To estimate Points F and H from the elevation anomalies … was calculated from the corrected …*"

**Lines 136-160 –** Following my general comment, these paragraphs are very difficult to follow, as a) they flick back and forth between the identification of *F* and *H*, and b) it is highly unclear what 'new' identification procedures the authors have used versus that discussed in Fricker & Padman (2006). There is also a lot of repetition of content already mentioned in the introduction. I strongly suggest re-structuring these paragraphs to closely follow the structure outlined below:

First paragraph: Very brief overview of the anatomy of the GZ with respect to Points *F* and *H* and the work of Fricker & Padman (1-2 sentences). Then, a few sentences on how \**they*\* identified the grounding zone.

Second paragraph: A discussion of how the authors expanded upon this earlier work and identified Point *F* using new slope-corrected techniques, MAEA and its second derivative. No mention of *H* here.

Third Paragraph: Discussion of how using MAEA alone is likely more difficult to identify Point *H*, and subsequently: how the authors used MAEA together with their error function to locate *H*. When discussing the error function, elaborate more on what "*as a guide*" means (readers should be able to replicate this method, so as written this seems vague/ambiguous).

**Line 163 –** unneeded '*the*' before elevation change. Also, this and next sentence are almost complete repetition, and should be condensed (and possibly merged with the following sentence). Also define '*KDTree*' (and any other abbreviation) in full when using for the first time.

**Sentence beginning line 168 –** Poor grammatical structure.

**Line 170 –** Should this say '*from*' instead of '*for*'?

**Line 175 –** Add Padman et al. citation after reference to the CATS2008a model. '*Movement*' should be singular.

**Line 178 –** '*based on the results in section 4*'. What does this mean? State the explicit steps you took to derive this number here, and then refer to the results section, if still necessary.

**Sentence beginning line 180 –** It is implicit in the above sentence that you did this, so suggest the removal of the sentence from the text.

**Line 183 –** suggest rewording to better match the structure of my suggested abstract changes. Suggest something like:

"*Using our newly developed ICESat-2-based GZ detection algorithm, we recover a two-fold increase in the number of GZ features identified over Larsen C Ice Shelf by ICESat-1 (69 and 71 picks of Points F and H, respectively, versus 30 of each by ICESat-1). The spatial distribution of each platform's GZ picks is shown in Figure 6, together with a comparison of Point F as determined from independent DInSAR observations. The improvement in our ability to image the GZ using ICESat-2 data is especially notable in heavily crevassed regions like*

*Jason Peninsula and Churchill Peninsula (Jansen et al. 2010), which were previously difficult to image using single-beam ICESat-1 observations alone*".

**Lines 189-195 –** As per my technical comments on Lines 183-216. Suggest restructuring this paragraph and Figure 5 to only make reference the more up-to-date ESA CCI product.

**Lines 189-201 –** This paragraph is another example of the somewhat unorganized writing style discussed above, this time pertaining to the back-and-forth comparison of different sensor observations. Mirroring my suggested edits to the methods section regarding the derivation of *F* and *H*, I suggest re-structuring this and the following paragraph in the following order to greatly improve clarity.

First para: Discussion of how the author's results compare with DInSAR observations (ESA CCI only). (Any comparison of ICESat-1 and -2 GZ points are largely unconvincing and should be excised from the text here).

Second para: Discussion of where your DInSAR and ICESat-2 *F* points don't agree (i.e. the para beginning line 202. Nice findings, by the way!).

Third para: Discussion of how your *F* points compare with those imaged by other sensors, including CryoSat-2 (nb. A discussion of CryoSat-2 vs. ICESat-1/DinSAR observations is outwith the confines of this study, and as a predominantly ICESat-2 methods study, should be removed. Also note the incorrect spelling of *CryoSat-2*). In accordance with my earlier comments, I think this paragraph would also be the place to add in comparison of your $I_b$ picks vs. those of the ASAID product).

Fourth Para: Discussion of Point *H*, encompassing my comments on lines 233-235 above.

**Lines 206-210 –** "*(vertical red dash line)*", "*(blue dash curve)*" etc. is non-essential information best placed in the caption instead.

**Lines 209-210 –** Highly verbose sentence, and repetitive use of 'left' and 'right'. This sentence should be condensed.

**Line 220 –** Insert comma before and after *|dh|*.

**Line 221 –** Change "*the repeat track analysis*" to "*our repeat track analysis*"

**Line 223 –** What does "*same time difference*" mean? Suggest rewording for clarity.

**Line 224 –** Suggest changing to "*The locations of Point F identified using our repeat-pass technique (black dots in Figure 8c) are located in the middle of high and low |dh| observations, indicating the presence of floating and grounded ice either side of Point F, respectively*".

**Paragraph beginning line 227 –** For simplicity and clarity, I would restructure this paragraph to first mention why we're interested in knowing the width of the GZ, then discuss how it has been calculated in the past and then how the authors have calculated it in this paper. Then finish with a discussion of your results. Similar to the discussion of Points *F* and *H* within the context of independent observations, some brief words on how measured GZ widths compare with e.g. ASAID GZ widths would also be well placed here. Also, the first sentence of this para is repeated directly from the introduction, so suggest rewording.

**Line 239 –** Suggest changing to: "*… for recovering several commonly imaged components of the ice-sheet-shelf grounding zone, including the inland limit of tidal flexure (Point F) and the limit of hydrostatic equilibrium (Point H). The new method presented in this study should …*"

***Line 240 –*** *"The new automated method presented in this study should allow a more efficient and consistent mapping of the Antarctic GZ features".* I see what the authors are trying to suggest here, but don't think it's articulated quite correctly. The authors should bear in mind that due to recent advances in polar-orbiting EO capabilities (incl. the launch of Sentinel-1a/b), we can in theory now generate a precise understanding of the GZ every 6 days using DInSAR. The spatial-temporal coverage of e.g. DInSAR-derived *F* will therefore *always* be better than ICESat-2 *F*, so I think this sentence (and any similar arguments made further down in the discussion and conclusion sections) should be re-worded to reflect this. What I think the authors want to say is that ICESat-2 provides an automated, efficient means of characterizing the Antarctic Ice Sheet's GZ, to complement the high-accuracy observations afforded by other modern Earth Observation sensors.

***Line 241 –*** This sentence should go into more detail about why we are interested in GZ position change and width (e.g. for improving numerical models aiming to constrain SLR predictions, identifying Antarctica's most vulnerable regions to climate change etc.).

***Line 244 –*** Suggest rewording to "*To maximize repeat-track coverage over the GZ, our method considers the ground tracks of one ICESat-2 repeat cycle beam pair as two individual repeat tracks*".

***Line 247 –*** Should say *"… which was an issue when using ICESat-1 data to identify the GZ (Brunt et al, 2010) …". Estimation* implies imprecision, so *identify* or similar is better".

***Line 248 –*** "*In addition, this also allows us to calculate the GZ features from just two time stamps of tidal amplitudes*". It's obvious to me why the authors say this, but to some this might imply an element of inaccuracy in the author's results. This is because recent research has shown that the GL can undergo significant changes in its position over a range of tidal frequencies, in some cases by as much as 4 km (e.g. Milillo et al., 2017, *GRL*).

I suggest the authors acknowledge this important point, and mention that while their technique does indeed work using only 2 repeat cycle's worth of data, the identification of longer-term GZ change completely distinct from tidal variability (e.g. Friedl et al. 2019) may only be possible as/when more ICESat-2 observations become available.

***Line 240 –*** see my comments re: Line 240. While this was perhaps once true, with the launch of Sentinel-1a/b (and soon -1c) I think the authors should reword this to state how the two datasets will *complement* each other.

***Paragraph beginning Line 257 –*** This paragraph is a bit 'matter-of-fact' or bullet-point-like, and at times not merited within the context of this paper (e.g. the review-like discussion of CS2 vs. ICESat-1 observations, which are independent of the method presented here). In any case, for simplicity I would reword the paragraph to echo the structure the results section (see my comments on Lines 189-201), and again excise any discussion of ICESat-2 vs. ICESat-1 precision from the text. As this is the discussion section, a regurgitation of the exact separation distances is also not essential here. Also, to avoid any possible confusion, I would refrain from referring to '*GL detection*' when referring to non-Point *G* components of the GZ.

***Line 264 –*** " *ice plains that are difficult to capture from DInSAR* …". Careful! Sentinel-1 DInSAR can capture it just fine (see below), so I'd reword this.

[Figure]

**Figure 1 –** Churchill Peninsula ice plain detected from double-difference InSAR applied to 2019 Sentinel-1a/b data.

***Line 268 –*** "*…, it demonstrates*". What demonstrates?

***Lines 270/271 –*** Where are these tidal estimates derived from? CATS2008a? Cite the appropriate reference here. Also, remove the space separating this and the next paragraph.

***Lines 279 –*** Suggest rewording to something more concise like "*Surface elevation changes derived from ICESat-2 cross-over data also provide valuable information about the approximate location of Antarctic grounding zones*".

***Line 281 –*** should say "… *separation afforded by repeat track observations*, *it can still reveal the approximate location of the GZ*". I also suggest the next sentence reads: "*In doing so, this method has the potential to provide important validation of repeat-track-derived GZ features, including along fast-flowing ice streams where the GZ often undergoes rapid changes …*" or similar.

***Line 283 –*** Restructure sentence to condense and avoid the use of '*Also…*'. This and the next sentence could also be easily merged.

***Line 288 –*** This sentence is rather verbose, and I think it could easily be shortened and/or restructured to convey the key selling point of paper more effectively (see my wording of the abstract above).

***Sentence beginning Line 288 –*** '*the* 8-month period'.

***Sentence beginning Line 293 –*** Larsen C Ice Shelf is a pronoun and so the preceding use of the word '*the*' is incorrect and should be removed. Next sentence once again begins with '*it*'. What does '*it*' refer to? Reword for clarity.

***Sentences beginning Line 294 –*** I would let the readers decide if the separation between your ICESat-2 and DInSAR observation are small, so remove '*small*' here. Also change all instances of '*GL*' to '*GZ*' for accuracy, and hyphenate all uses of the word "*derived*".

***Line 296 –*** Again, I'm just not sure you can confidently talk about ICESat-2 vs. ICESat-1 precision that way. Also, as far as I can tell, in the current version of the manuscript the authors make no direct comparison between their ICESat-2- and any CryoSat-2-derived *F*

products, so don't think any robust statements can be made regarding precision in that sense either. As a result, I think this sentence should be restructured to echo my suggested restructuring of the results section.

**Line 297 –** Change '*this method*' to '*our method*' and remove '*the*' before Larsen C Ice Shelf.

**Line 298 –** Insert a comma after '*range*'.

**Line 299 –** rephrase to avoid "*it still shows*". Also '*GL*' should say '*GZ*'.

**Line 301 –** Suggest changing to "*…Antarctic Ice Sheet, similar cross-over-based analyses have the potential to provide more accurate depictions of the GZ*".

**Sentence beginning line 301 –** This is almost complete repetition of lines 293-295, so I suggest removing it from the text.

**Line 310** – Where can the ESA CCI DInSAR data be found?

**Line 333 –** Incorrect dataset cited.

**Figure 1** – It may just be the way the PDF has rendered the image, but as printed, the vertical lines pointing to each of the GZ components have different line thicknesses and slightly different colors. Suggest the authors standardize all thicknesses and colors if this is not a rendering issue.

**Figure 1 caption** – Capitalize all instances of '*Point*'.

**Figure 2 –** As previously discussed, suggest the authors clip their dataset to the modern-day coastline (i.e., account for A-68's calving where elevation clearly equals zero over open water in the fig) and only show the 2017 (ESA CCI) grounding line. I also wonder if the authors inadvertently clipped Bawden Ice Rise out of their analyses? Like Gipps Ice Rise, this is an important pinning point for Larsen C (c.f. Borstad et al., 2017; *GRL*), so it'd be interesting to see any available GZ picks here. I suggest also standardizing the font size of all labels on the main map, and instead assigning track labels a different font color to better contrast against place-names. I also suggest center justifying the scale bar relative to the other items of the legend, and moving all items slightly to the left (at the moment the 1000 m label intercepts the figure border and looks untidy). Taking potentially color-blind readers into consideration, I'd also encourage the authors to choose another, non-jet/rainbow color ramp.

**Figure 2 caption** – Capitalize '*Ice Shelf*' and amend caption to reflect the addition of the 2017 (ESA CCI) grounding line. Remove '*in Antarctica*' as the inset clearly shows this. As the grid is shown in meters, please also state the coordinate reference system used (e.g. EPSG:3031 Antarctic Polar Stereographic) (same for Figs. 6 and 8).

**Figure 3** – As per my comments on the paragraph beginning Line 93, authors might consider enclosing the various sub-sections of their methodology with boxes. '*Nominal Reference Track*' and '*Hydrostatic Point H*' should also be shifted so as to not impinge on the boxes. Capitalize '*Point F*' and '*Point H*'.

**Figure 3 caption** – Suggest "*The ICESat-2 repeat-track workflow used to identify the grounding zone (GZ) in this study, including the limits of inland tidal flexure (Point F) and hydrostatic equilibrium (Point H)*" (note: In general, abbreviations should be written in full in figure captions, which is why I have done so here).

**Figure 4 caption** – Suggest either amending the figure or stressing in the caption that this image only refers to 1 of the 3 beam pairs imaged by ICESat-2 at any one time (not all readers will be familiar with the 3-beam imaging capabilities of ICESat-2).

**Figure 5** – Why is repeat cycle 5 not shown? In the main text it is stated that this dataset is used, so its omission here is rather odd. Suggest also ordering the repeat cycle legend in numerical order (*3l, 3r, 4l, 4r* and so on)

Also, I don't quite understand the inclusion of elevation by latitude (dashed profiles) here? The only reason I can think of is because track 1192 overpasses two GZs on the north and south of the unnamed ice rise seen in Fig.2 (and so should have some easily identifiable break-in-slope), but even then it's unclear where this happens. In any case, when superimposed over the anomalies and other info it makes for a rather busy diagram, so I'd suggest removing it for simplicity. If the authors feel this is essential information, then I recommend they include it either as an additional sub-panel to this figure (cf. Fricker et al., 2006) or as a supplement.

In all panels, some of the axis and other labels appear cut-off, so care should be taken to fix these minor blips.

**Figure 5 caption** – This is a very long and difficult to comprehend sentence (and caption overall). Can it be condensed somehow? I'm also not sure the second sentence makes grammatical sense, and could be shortened to: "*in the legend, c03_1r refers to the right ground track of ICESat-2 pair 1, repeat pass 3*".

After the Padman reference, insert "*following Fricker & Padman, 2006*". Information about the derivation of MAEA has already been discussed in the text, and so should not be included in the caption. Capitalize all instances of the word '*Point*'.

**Figure 6** – Same comments as Figure 2 regarding the center justification and movement of the items in each of the legends. Authors may also wish to consider clipping the velocity map for consistency with Fig. 2. Update DInSAR coverage to only show the 2017 Point *F*. In *c* what do boxes *A* and *B* refer to? (This isn't stated in the caption). I think they refer to the spatial extent of Figs. 8a/b - if so, suggest labelling '*Fig. 8a*' and '*Fig. 8b*' here, and similarly indicating the extent of Fig. 8c. Also standardize the thickness of the DInSAR Point *F* line in all plots.

**Figure 6 caption** – Once again, '*estimates*' inherently suggests imprecision, so suggest rewriting caption to begin: "*ICESat-2-derived inland limits of tidal flexure (Point F; red dots). For comparison, ICESat-1-derived Point F (Brunt et al., 2010) picks are also shown (blue dots), and data are superimposed over on recent ice surface velocity magnitudes (Rignot et al., 2017). b) same as a), but showing …*". Then, at the end, "*in all plots, DInSAR derived Point F is also shown*".

**Figure 7** – As per my comments on Fig 5. Also, is there any way to re-size the labels in *c* and *d* so that they don't cross over the reference lines and data? At present these look messy.

**Figure 7 caption** – Another long and verbose caption which, being very similar to the caption of Figure 5, can be easily shortened to "*Same as Figure 5, but showing …*".

**Figure 8** – In all subplots, show only the 2017 ESA CCI Point *F* lines. In *c*, a black circle appears to be obscuring a blue circle. Suggest using unfilled black circles (i.e. outline only) in all subplots to avoid this. Suggest also scaling the black circle in the legend to match the its size as shown in the sub plots.

**Figure 8 caption –** Incorrect grammar. Suggest "*Spatial distribution of ICESat-2 crossovers analyzed in this study. The spatial extent of plots a-c is shown in Figure 6c. At each location, the absolute change in elevation, |dh|, is shown. Background is …*".

**References cited in this review**

Adusumilli et al., 2017, *Geophysical Research Letters*, doi:10.1002/2017GL076652

Bindschadler et al., 2011, *The Cryosphere*, https://doi.org/10.5194/tc-5-569-2011

Borstad , McGrath and Pope, 2017, *Geophysical Research Letters, https://doi.org/10.1002/2017GL072648*

Fricker et al., 2012, *Journal of Geophysical Research: Oceans*, doi:10.1029/2011JC007126, 2012

Scambos et al., 2007, *Remote Sensing of Environment*, https://doi.org/10.1016/j.rse.2006.12.020

Shepherd & the IMBIE team, 2018, *Nature*, https://doi.org/10.1038/s41586-018-0179-y

---

## Referee Comment (RC2) · Anonymous Referee #2 · 26 Jun 2020

Mapping the Antarctic Grounding Zone from ICESat-2 Laser Altimetry
Tian Li, Geoffrey J. Dawson, Stephen J. Chuter, Jonathan L. Bamber

This paper uses ICESat-2 data to identify the grounding zone of the Larsen C Ice Shelf, Antarctica. It builds off of methods developed for ICESat and automates some of these processes. The authors describe a modified repeat-track analysis (that uses 6.5 months of ICESat-2 data) and a crossover analysis (that uses 1 year of data) to determine a GZ that is in close proximity to the continuous GL assessment from the DInSAR GL of Rignot et al., 2016; this agreement is heavily based on the increased resolution of the ICESat-2 data, and not a measure of precision, as the authors contest.

The paper overall needs a fair bit of rewriting, with some text being repetitive and other sections feeling out of place. More care needs to be taken with respect to using uniform language with respect to terms that are introduced. And more care needs to be taken with respect t to references ('et al.' is often, but not always, missing).

The method presented here builds off of methods developed for ICESat (please change all references of 'ICESat-1' to 'ICESat'). Some of the modifications may be relevant (such as the across-track slope correction) but are not strongly justified; other changes (or additions) do not seem to add to the results on this ice shelf (such as the crossover analysis).

My major comments include:

1) Why do the authors utilize a across-track slope correction and migrate the analysis to the 'nominal reference track'? In line 97, the authors state that the ICESat-2 "separation between same side ground tracks in a pair from different repeat cycles is less than 10 m". I do not expect substantial across-track slope, or rough terrain, at 10-m length scales. Uncorrected elevations in Figures 5a and 5b confirm this. The authors could instead produce repeat-track assessments through the GZ for all 6 ground tracks, increasing the number of point F and point H assessments substantially. If the across-track slope correction is needed for more subtle reasons, the authors need to better justify this.

2) I do not believe that the crossover analysis provides any additional merit to the existing repeat-track analysis for the Larsen C. As the authors mention, it might be more useful in more southerly ice shelves. I suggest removing that element from the paper and using it at a later date.

3) The authors state that the agreement between this GZ and the GL of DInSAR is an indicator of increased precision, relative to ICESat; 'precision' is not a great term for this. The close agreement is undoubtedly due to the increased across-track and along-track data density associated with ICESat-2; this is a spatial resolution improvement, not a 'precision' improvement.

Specific comments:

Throughout: 'ICESat', not 'ICESat-1'
Throughout: Clean up the references (missing 'et al.' often)

Title: this is a bit misleading, suggesting that this is a result for the entire Antarctic. The definite article is not appropriate here. I suggest somehow adding 'Larsen C' to the title.

Line 10: the results *should* show a three-fold increase; why only 2?

Line 30: Point I is poorly characterized in Fig 1 and not mentioned in this paragraph. Suggest removing it from figure.

Lines 49 – 51: "The launch of ICESat-2 on 15 September 2018, as a successor to the ICESat-1 satellite mission, can achieve higher along track spatial resolution as well as nine times better coverage compared with its predecessor, providing the potential to map the GZ with greater accuracy and spatio-temporal coverage."
This needs to be better described. There are both along-track and across-track data density issues here. And there are more repeats within a calendar year. But that's not captured explicitly in this text.

Lines 64 – 68: The time span is unnecessarily complicated in this section. A couple options: 1) remove the crossover analysis. 2) reduce the time span of the crossover analysis to match the repeat-track analysis.

Lines 67 – 68: Note that ICESat-2 was not operational for much of July. And the analysis stops halfway through November. Thus, there are really only 6.5 months of repeat track analysis, and cycle 4 (?) is not a complete cycle.

Line 69: "… ICESat-2 did not point at the Reference Pair Track (RPT)" Careful here; ICESat-2 points at the RGT; the RPTs are specific to the ATL06 data product.

Line 73: The start of this paragraph could be smoother; something like, ICESat data were corrected for ocean tides and had to be retided; ICESat-2 ATL06 data are not…

Lines 81 – 82: "In addition, the surface elevation was also compared with the reference DEM Reference Elevation Model of Antarctica (REMA) (Howat et al. 2019) provided in the ATL06 product…" This is extremely challenging in the region of the GZ. REMA elevations are not tide corrected; thus, their elevations in the GZ are time/scene dependent.

Line 83: "elevation differences with REMA fell outside a ±150 m threshold"
What? 150 m of elevation differences?

Line 108: "300 m were removed as the surface elevation of GZ"

How many points did this remove? Were many of these points picked up by the ATL06 quality summary?

Line 112: "The elevation change calculated from repeat tracks is a combination of cross-track slope induced elevation change …"
You say above that the repeatability is 10 m; I do not think that there is appreciable slope on 10-m length scales. I think what you are trying to say is that 1) you are prescribing height changes to the Nominal Reference Track (language from figure) or the RPT (language from above); 2) slope on 90 m length scales are important. Suggest 2 things here: 1) get the language between figure 4 and the text above more common; and 2) state from the onset that your analysis is being done on the RPT. Although, as I stated above, I do not think a slope correction is needed with 10 m repeatability.

Lines 113 – 114: "As only 8 months of repeat cycles from March to November 2019 were used, …" As stated above, it's really only 6.5 months with the loss of July.

Line 121: Define these terms before equation 2 is introduced. Spell out $h_{LI}$ is (I assume) height for 'land ice'; but what is $y_{ATC}$? What does ATC stand for?

Line 125: RGT? Or the actual surveyed ground track?

Line 126: Nominal? Or calculated mid-point for the actual surveyed ground tracks?

Line 126: "The average of all cross-track slope corrected elevations…" I'd be clearer and uniform with language. For clarity, start with 'elevations' (that's what they are) and then use a single term throughout (e.g., the start of the next paragraph); I'd call these 'elevations corrected for cross-track slope'.

Line 129: "After the cross-track slope correction for each ground track in a pair, …" This is not clear.

Line 130: 'reference elevation profile …' Use uniform language when defining terms; make this match the language from the last paragraph.

Lines 131 – 132: Watch the order of how you introduce Figures.

Line 141: "consistent with the local tidal amplitude…" Cite the source of that amplitude.

Line 152: This paragraph needs a rewrite; it's overall confusing

Line 158: "predictions at 10 km offshore…" 10 km may be too large to extend the methods to complex GZs; suggest 1 km.

Line 163: The sentence starting on this line is confusing. I think the authors are using the KDTree to get data in proximity to crossover points, and then finding the absolute location using the quadratic functions? This needs to be cleaner/clearer.

Line 164: Wrote out and partially describe KDTrees.

Line 168: What is 'THE valid elevation measurements'?

Line 173: "where the elevation exceeded 20 m": It's totally fair to have a cutoff; but suggesting that 20 m is a temporal (accumulation?) change is not the right justification.

Line 175: cite CATS.

Line 177: The sentence starting on this line isn't clear to me; I think you used a 40 cm offset for the crossovers based on 20 cm offsets from the repeat-track analysis. But it needs to be cleaner/clearer.

Lines 184 – 191: Every reference in this section is missing an 'et al.'; this is common throughout. I'm not looking for all of them.

Line 192: "ESA CCI DInSAR GL (ESA Antarctic Ice Sheet Climate Change Initiative, 2017)" Delete the abbreviation part of this.

Line 192: How is 'separation' defined? Absolute distance? Distance along the ground tracks?

Line 199: For the sentence that starts here, this really isn't a validation; these results just suggest that the point H identified using the repeat-track method is actually on the fully hydrostatic part of the ice shelf; but it doesn't validate that it's the first point of hydrostatic equilibrium, or point H.

Lines 206 – 209: I am not sure what this is saying.

Lines 206 – 216: I do not follow the line of thought here and therefore am not convinced that this rationale and Figure 7 suggest an ice plain.

Line 215: 'Coupling Line' is not fully defined; the authors should also cite Hugh Corr's work on Pine Island Glacier.

Line 219: " … crossover analysis can provide additional information on the GL location." I don't think that this is true for Larsen C; there isn't the spatial resolution for this to add (or validate) the locations of points F and H. I suggest saving this method (crossover analysis) for a later paper, where the method is more appropriate.

Lines 225 – 226: I do not think that this method provides a validation. What about the pink point above the black F in 8c?

Line 231: "By assuming that the reference GL used in section 3.1 …" Cite Depoorter again to remind people what you are referring to.

Line 234: '9.73 km': Why such a large outlier?

Lines 238 – 240: "The aim of this study is to assess the capability and the accuracy of ICESat-2 laser altimetry data for estimating the GZ features of the Antarctic Ice Sheet including flexural point F and hydrostatic point H." I'd remove this; it doesn't offer anything new. But the next sentence does.

Lines 244 – 245: "… the new method takes two ground tracks of one beam pair from one repeat cycle as two repeat tracks." I do not understand this sentence.

Lines 253 – 254: "…ICESat-1 repeat track analysis which required multiple repeat tracks (normally from at least five repeat cycles) with large temporal differences (Brunt et al. 2010; Brunt et al. 2011)." This statement is not accurate and was not a result in the references provided. There are figures in those papers that use fewer repeats.

Line 258: (and throughout) As noted in my initial comments, 'precision' is not a great term for this. The close agreement is undoubtedly due to the increased across-track and along-track data density associated with ICESat-2; this is a spatial resolution improvement, not a precision improvement.

Line 267: 'and'; the 2 clauses on either side of this have nothing to do with one another.

Line 272: The results in this paragraph are a direct repeat of the results from Fricker & Padman 2006, and Brunt et al., 2010, 2011. The authors need to present how they have added to these results.

Lines 279 – 280: "provide valuable information about the position of GL in the Antarctic Ice Sheet." I don't think this is accurate. I don't see what these assessments add, with respect to location of the GZ.

Line 281: "it still can show the general location of the GL" I just don't see this.

Figure 1: Consider removing point I or including it in the text with detail. From the figure, it is hard to discern if this is capturing I_m or I_b.

Figure 4b: I would not repeat the color scheme of 4a; this suggests that different cycles are required for this method.

Figure 5: Trim the caption. For the legend next to 5b: why are they ordered like this? Why are the purple and olive lines not included? What are the purple and olive lines?? I think they are elevations. It looks like there are 4 lines there (blue and orange??). That is confusing.

Figure 7: Trim the caption. Some of the language (e.g., defining MAEA again) is repetitive and can be removed. Overall, this figure is confusing and does not convince me that this is an ice plain. Figure 7d is especially confusing.

Figure 8: 8b shows that the spatial resolution of these crossovers does not add to the results associated with the repeat-track analysis. Suggest removing this.

---

## Author Comment (AC1) · 6 Aug 2020

**Author response by Tian Li on behalf of all the other authors:**

We would like to thank the reviewers for their exceptionally detailed and constructive comments on our paper, which have proven very helpful in revising this manuscript. We have been able to fully address the suggested changes and we think the revised manuscript has been greatly improved. Our detailed responses to each comment presented by the reviewers are in blue text.

**Reviewer 1**

**Review of "Mapping the Antarctic Grounding Zone from ICESat-2 Laser Altimetry" by Tian Li et al.**

Li et al. use 12 months of observations from the newly launched ICESat-2 laser altimeter to recover two key components of Larsen C Ice Shelf's grounding zone (GZ). Building upon the repeat-track methodology first applied to ICESat GLAS data (e.g. Fricker & Padman, 2006; Fricker et al., 2009; Brunt et al., 2010; 2011), the authors use elevation departures from mean tidal state to infer the landward limit of tidal flexure and point of hydrostatic equilibrium over successive repeat passes. By exploiting the beam-pair imaging capabilities of the ATLAS instrument on-board ICESat-2, the authors also apply a cross-track slope correction to their dataset, greatly improving the signal-to-noise ratio of GZ detection over areas of rough terrain compared with single-beam ICESat methods. Elevation changes derived from ICESat-2 ascending and descending cross-over passes are also used to approximate the position of the GZ (although at much degraded resolution c/w their repeat-track observations), thereby acting as a part-validation of their repeat-track-derived GZ picks.

Given the close agreement between ICESat-2 and independent InSAR-based GZ observations (~+/- 0.5 km separation, assuming no GL migration between acquisition dates) and the fact that this technique is fully automated, I believe that Li et al's technique will be of large interest to the readership of *The Cryosphere*, and look forward to seeing this paper published. The prospect of applying this technique to the entirety of Antarctica's grounding zone is also an exciting one. Before acceptance, however, I feel that moderate revisions must be implemented to improve the clarity and flow of the manuscript, and to address several technical points pertaining to the GZ detection technique and comparison with other data sources. My suggested revisions and other comments are included below.

This is a very good summary for the methods and results presented in our study and we would like to thank the reviewer for the detailed comments. We have now made the following changes to address the major comments from both the reviewers.

i. We used the latest ATL06 version 3 data during a 10-month period between March 2019 to March 2020, including four repeat cycles (3, 4, 5 and 6) for both repeat track analysis and crossover analysis. The GZ results have been recalculated over the Larsen C Ice Shelf.

ii. We have modified the reference GL used in repeat track analysis by merging the 2015-2016 ESA CCI DInSAR Point F (ESA Antarctic Ice Sheets CCI, 2017) and the Depoorter et al. (2013) GL.

iii. We added the repeat track analysis for six individual ground tracks in addition to the repeat track analysis for beam pairs used in the previous submission. Although we do not expect a substantial across-track slope over a ~10 m separation for repeat tracks of each ground track, using beam pairs with the across-track slope correction is essential for high-sloping regions. As the example shown in Fig. 6, the ~10 m across-track separation can still lead to significant across-track slope induced errors in elevation change (Figs. 6a-6d), but the across-track slope correction can reduce this error (Figs. 6e-6h).

iv.   We have modified the crossover analysis by using the average elevation change at each crossover location, this removes a lot of noise observed in the previous data.

v.    We only compared the ICESat-2-derived Point F with ESA CCI DInSAR Point F, and removed the comparison between ICESat-2-derived Point F with Rignot et al. (2016) DInSAR GL and the ICESat-derived Point F (Brunt et al., 2010). We have also compared the ICESat-2-derived Point H with ASAID Point H (Bindschadler et al., 2011).

vi.   We have modified the overall structure and the clarity of the paper, especially Sections 3 (Methodology) and 4 (Results and discussion).

**General / Major Comments**

While it's clear that the authors have invested a large amount of time and effort in the development of their GZ detection technique, I feel that large parts of the manuscript should be thoroughly edited by all authors to increase the overall clarity, flow and standard of English presented in the text. At present, the main text is at times highly verbose and/or repetitive, which – together with the use of non-standard sentence/paragraph structure throughout the manuscript – makes it difficult for the reader to easily follow the narrative of the paper. This is especially true in the methodology and results sections, where, for example, the reader must repeatedly flick back and forth between the discussion of Points F and H - this has the potential to be extremely confusing, especially to those unfamiliar with the anatomy of the GZ. As I outline in my specific comments, these sections could easily be restructured to offer a more streamlined discussion. I would also strongly encourage the authors to correct the numerous instances of incorrect grammar contained in their manuscript (I have indicated some, but not all instances of this in my specific comments).

We agree with the reviewer that we need to improve the overall clarity of the context. We have substantially modified the structure and language based on reviewers' comments and we believe the overall clarity, flow and the standard of English have now been greatly improved.

***Lines 183-216 –*** To my knowledge, the 2017 ESA CCI dataset provides near-complete coverage of Larsen C's GL as constrained from Sentinel-1a/b DInSAR-based observations acquired in 2016/2017. I am therefore surprised to read that the majority of the ICESat-2- DInSAR comparisons made in this manuscript refer to the earlier (mostly 1990s ERS-1/2- based) GL products provided by Rignot et al., (2016), and not those more temporally contemporaneous with the authors' ICESat-2 observations.

We agree with the reviewer that it will be more useful to compare our results with the most up-to-date ESA CCI grounding line product. We have now removed the comparison between ICESat-2-derived Point F with the DInSAR grounding line provided by Rignot et al. (2016), and only compared our results with the ESA CCI DInSAR Point F. Please refer to line 12-15, line 276-288, line 360-362, Table 2, Figs. 7 and 10.

Furthermore, while the authors correctly mention in lines 106/107 that recent (2010-2016) GL migration has been fairly minimal, a comparison of the Rignot and ESA CCI products reveal large retreats (up to 4 km) along parts of Larsen C's GL (including Churchill and Kenyon Peninsulas), presumably in response to the sustained period of ice-thinning between 1979 and 2008 noted in the literature (Fricker et al., 2012, JGR: Oceans; Adusumilli et al., 2018, GRL). (NB. Since 2008, Larsen C has largely thickened, consistent with the more stable rates of GL migration reported by Konrad et al. (2018)).

This is a very good point. We use the fact that the grounding line on Larsen C Ice Shelf has been relatively stable during our observation period for two reasons. The first is for the comparison with DInSAR grounding line observations. Especially as we are now using the latest ESA CCI grounding line

product for comparison with the ICESat-2-derived Point F, we are confident that the grounding line migration has been very small between the two measurements. The second is to define a calculation window for our automated grounding zone detection algorithm. The 24 km calculation window for the reference GL, which was the Depoorter et al. (2013) GL in the initial submission, should be sufficient to capture the grounding zone within it, even including the ~4 km GL retreat between 2010 and 2016. Nevertheless, to avoid the uncertainty in the location of the Depoorter et al. (2013) GL which was derived from a composite of outdated GL products, we have now modified the reference GL by merging the 2015-2016 ESA CCI DInSAR Point F and the Depoorter et al. (2013) GL (line 120-129).

As such, I believe making primary reference to the old Rignot GL dataset is inappropriate in this context, and would strongly suggest the modification of the abstract, lines 183-216, discussion/conclusion and Figures 2, 5, 6 and 7 to refer only to the newer ESA CCI GL. Also, be sure to include the appropriate citation for the ESA CCI dataset in the reference list.

Agree, we have now removed all the comparison with the Rignot DInSAR GL dataset and only compared ICESat-2-derived Point F with ESA CCI DInSAR GL. We have changed the correct citation for the ESA CCI dataset in the reference list into 'ESA: Antarctic Ice Sheet Climate Change Initiative, Grounding Line Locations for the Fleming and Larsen Glaciers, Antarctica, 1995-2016, v1.0, Centre for Environmental Data Analysis, 2017.'

***Technical Comments***

***Line 71 and Figs 5 and 7*** – The authors mention the use of data from repeat cycles 3-5 in their repeat track analyses, but Figures 5 and 7 only show the solution as derived from repeat tracks 3 and 4. Why is this? For consistency with the main text, I'd update Figures 5 and 7 to show the full complement of data acquired over the entire range of images tidal states.

Agree, we have modified the Fig. 5 to include all the repeat cycles (3, 4, 5 and 6) used in this study to show the full tidal range.

***Line 102 –*** As the authors know, the Depoorter GL is a compilation of multiple GL datasets, derived from a variety of techniques (e.g. InSAR, break-in-slope, ICESat) applied to satellite imagery acquired at different times. For absolute clarity here, I'd consider adding an overview of the imaging times and techniques used over this region of Antarctica.

Agree and done. We have modified the reference GL by merging the latest ESA CCI DInSAR GL (2015-2016) with the Depoorter et al. (2013) GL to avoid the uncertainty of the GL location in the calculation (line 120-129). An overview of the imaging times and techniques used in Depoorter et al. (2013) GL has also been added in line 120-129.

***Lines 120-125 –*** For clarity, I suggest defining each of the terms included in Eq. 1 before discussing Eq. 2. Also note the lack of any explicit definition of $Y_{ATC,G}$ in the current version of the manuscript.

Agree and done. $Y_{ATC,G}$ is the y coordinate measured perpendicular to the Reference Ground Track (RGT). For clarity, we have now changed $Y_{ATC,G}$ to $Y_L$ (y coordinate measured perpendicular to the RGT for the left beam in a pair) to demonstrate the across-track slope correction from left beam to the nominal reference track in the middle of the beam pair (Eq. 1 and 2).

***Lines 136-160 –*** In Figure 1 the authors show the position of $I_b$, the ice-sheet-shelf inflexion point (otherwise known as the break-in-slope), which is obtainable from optical-, DEM- and altimetry-based

methods. As the authors are aware, $I_b$ has been used extensively in the literature (including Christie et al. (2016) as mentioned in the text, but also e.g. Bindschadler et al. (2011, *The Cryosphere*) and Scambos (2007, *Remote Sensing of Environment*)) to supplement our understanding of the GZ in areas where DInSAR coverage is poor. I was therefore a little surprised to see no discussion or method of deriving this important GZ component using ICESat-2 in the text. This is despite the break-in-slope being clearly visible in e.g. Figures 5a and 5c. Was any attempt to automate this component of the GZ made? If not, why not? I think a brief discussion of $I_b$, its manifestation in the ICESat-2 data and any attempts to automate it should therefore be added to the Section 3.1. If easily obtainable, then an additional figure similar to Figure 5c could also be added to show any differences in ICESat- 2-derived $I_b$ and the break-in-slope mapped by ICESat-1 (Brunt et al., 2011) and/or the ASAID team (Bindschadler et al. 2011).

We have now explained the definition of Point I in line 33 and the previous research on identifying Point I in line 48-50. We agree that the break-in-slope can provide reliable representation for the grounding line in slow-moving regions, but this method is less effective in identifying fast flowing glaciers with low surface slope (Bindschadler et al., 2011; Scambos et al., 2007). The identification of Point F based on tidal flexure, however, is not influenced by these limitations. Also, the method developed in this study is based on detecting the changing points from the elevation anomalies rather than the surface slope, this cannot be directly applied to the identification of Point I. While our method could potentially be modified to detect Point I, it is beyond the research scope of this study.

On a related note, I have noticed occasional large errors in the positioning of DInSAR-derived GLs (Rignot, ESA CCI) over the Antarctic Peninsula (including Larsen C), presumably related to the geo-coding of SAR data over the Peninsula's steep terrain using old DEMs, and/or the misattribution of the Point *F* on ice landward of nunataks or other mountainous exposures at the GZ. These positional errors are not seen in e.g. the ASAID product due to the on-nadir viewing angle of Landsat and the ability to distinguish between mountains and ice more easily. Therefore, any comparison of ICESat-2- and ASAID-derived $I_b$ picks may be more representative of the true GL (in lieu of Point *F*) in these areas. (As a relatively slow flowing region with a well-defined break-in-slope, $I_b$ will fall close to Point *G* (as it does in Fig. 5), making it reliable proxy for Point *G* and *F*).

In light of the above and if not already done, the authors should thoroughly QC the DInSAR product for such offsets – their omission could actually improve your mean absolute separation statistics!

We have now compared our ICESat-2-derived Point F with ASAID break-in-slope point (line 282-288) and added a new figure (Fig. A1 in Appendix A) to show the spatial distribution of absolute separation. The result shows a close agreement between the two products, with an absolute mean separation of 0.34 km and a standard deviation of 0.28 km (Table 2). The overall separations in Churchill and Kenyon Peninsula are smaller than the DInSAR-derived Point F (Fig. 7c).

[Figure]

Figure A1. Absolute separations between ICESat-2-derived inland limit of tidal flexure (Point F) and ASAID break-in-slope point (Point I). ASAID Point I is shown as the black line (Bindschadler et al., 2011). Background is the recent ice surface velocity magnitudes (Rignot et al., 2017) in Antarctic Polar Stereographic (epsg:3031) projection.

**Lines 150 and 156** – How sensitive is the pinpointing of Point *H* to the prescribed variance threshold used in the error function, and why was the fourth derivative of this function chosen? In other words, how would e.g. a 3$^{rd}$ order fit look on Figure 5d, and where would *H* be placed as a result? Some discussion of your choice is needed in the text here.

We have now discussed the reason of choosing the fourth derivative of the Error function as a guide point in searching for Point H in line 179-184 and Fig. 5d. The vertical dashed grey line in Fig. 5d shows the position of Point H if using the same guide point for Point F (third derivative of the Error function). It is clear that this location is slightly landward compared to the actual Point H signal (the vertical dashed blue line).

**Line 190** – With the exception Churchill Peninsula (Figure 6a) where *one* ICESat-1 *F* point falls very close to the ICESat-2 *F*, spatially coincident ICESat-ICESat-2 observations do not exist anywhere else along the Larsen C coastline. I would therefore excise any comparison between the two datasets from the text as I don't think such statements can be made with confidence. Instead, I'd solely report on the close agreement between ICESat-2 and DInSAR observations. (the discussion of the enhanced coverage between ICESat-1 and ICESat-2 is okay, however!).

Agree and done. We have now removed all the spatial comparison between the GZs derived from ICESat and ICESat-2, and only kept the discussion on the enhanced density of ICESat-2-derived GZs compared to ICESat in line 249-251.

**Sentence beginning line 196** – The GZ information contained in Dawson and Bamber (2017) refers to Siple Coast, not Larsen C. Are the reported separation values mentioned in the text derived from those observations over the Siple coastline, or has the methodology of Dawson and Bamber been extended to Larsen C? This is essential information to state here.

The reviewer is correct that the GZ information contained in Dawson and Bamber (2017) only refers to Siple Coast on Ross Ice Shelf rather than Larsen C Ice Shelf. However, they have extended their methodology to Larsen C Ice Shelf in a most recent paper (Dawson and Bamber, 2020), and reported a standard deviation of 1.2 km between CryoSat-2-derived Point F and ESA CCI DInSAR Point F. Therefore, we have modified this argument to only include the comparison in the same region by Dawson and Bamber (2020) in line 279-280.

*Lines 217-226* – In Figure 8 (b and c especially), I see numerous instances of dark blues over the ice shelf which, following the main text, readers might incorrectly interpret as grounded ice. The opposite is true over grounded ice, which may be interpreted to be floating. What is the explanation for these observations? Surface processes? A few sentences explaining these blips are essential here.

Agree. For the crossover method, we now use a 10-month period of ATL06 version 3 data from 30 March 2019 to 6 March 2020 in order to be consistent with the repeat track analysis (line 65-70). We have also modified the crossover analysis by using the average elevation change at each crossover location (line 245-246), this removes a lot of noise observed in the previous data. To demonstrate the advantage and potential of crossover analysis, we have modified the figure to include the distribution of crossovers |dh| on Hearst Island (now Fig. 9), where the crossovers show a very obvious transition from land ice to floating ice (line 305-319).

*Lines 233-235* – Similar to my comments on the potential derivation of $I_b$ from ICESat-2 data and its comparison with e.g. the ASAID product, I am surprised to see no comparison of the authors results with ASAID-derived $H$. At present this our best pan-ice sheet understanding of $H$, so it would be very useful to see a discussion of how your results compare here.

This is a very good advice. We have now compared the separations between ICESat-2-derived Point H and ASAID Point H in line 320-326, Table 2, Figs. 7b and 7d. The comparison shows an overall good agreement, the absolute mean separation and the standard deviation between the two products are 1.2 km and 0.98 km, respectively. 83% of the ICESat-2-derived Point H locate less than 2 km away from ASAID Point H which are within the geolocation error of ASAID Point H indicated by Bindschadler et al. (2011).

*Line 275* – As no reference is included to support this assertion, the discussion of crevasses and their influence of GZ reads like something of an afterthought, potentially degrading the reader's confidence in this otherwise highly impressive technique. Since this is predominantly a methods paper, I'd encourage the authors to elaborate on a) why they didn't (couldn't?) clean up these data, b) exactly how they think these phenomena may influence the overall accuracy of their technique and, if needed, c) briefly mention how future work may address this issue. I imagine a lot of this could (and should) go into Section 3 so that the reader can bear these uncertainties in mind while reading Section 4. In addition, it is generally poor grammatical form to begin new sentences with 'Also', 'However', 'It' etc., so the readers should consider revising the structure of the next sentence (and several other similar instances throughout the manuscript) to address these minor niggles.

Agree. We have now added a new section 3.1.4 'Filtering and visual validation' to explicitly explain how we filtered the wrong GZ identifications due to inaccurate error function fitting or data deficiency. We firstly applied quality check flags in our algorithm to filter the GZ results:

   i.    Quality-0 : Potential good results passing the two quality checks below.
   ii.   Quality-1 : data loss percentage flag.  If the percentage of segments on the nominal reference track where there is no elevation measurement is larger than 50%, then it means this repeat-track data group does not have enough data to perform a reliable GZ calculation. This data deficiency can be caused by the cloud or snow.

iii.    Quality-2: inaccurate Error function fitting flag. Large across-track slope and crevasses can introduce significant amount of noise in MAEA curve and can result in an inaccurate Error function fitting when identifying GZ. If the distance between Point F and the reference GL exceeds 5 km, then the Error function fitting is possibly inaccurate.

However, there are several circumstances where quality flags are inaccurate in these three categories. For example, the existence of ice plain can result in up to 10 km separation between Point F and the reference GL, but the GZ results in this case are marked with 'Quality-2'. Also, large across-track slope can cause inaccurate GZ identification for single beam repeat-track data group (Figs. 6a-6d) in category 'Quality-0'. Therefore, we applied a final visual validation on all the GZ results with the aid of the flags to remove these wrong classifications.

***Line 277 –*** Good point. Does the processing chain provide a means in which the user can obtain the number of repeat-track observations your GL picks are based on? Such information could be used to calculate a 'GZ pick reliability rating', and/or inform data users about the range of tidal frequencies the GZ is mapped over.

This is a very good advice. We agree that both the number of repeat cycles used in GZ calculation and the tidal amplitude range are very valuable information for analysing the reliability of ICESat-2-derived GZ features, studying the short-term GL migration, or separating the long-term GL retreat signal as another comment made by the reviewer:

***Line 248 –*** *"In addition, this also allows us to calculate the GZ features from just two time stamps of tidal amplitudes". It's obvious to me why the authors say this, but to some this might imply an element of inaccuracy in the author's results. This is because recent research has shown that the GL can undergo significant changes in its position over a range of tidal frequencies, in some cases by as much as 4 km (e.g. Milillo et al., 2017, GRL). I suggest the authors acknowledge this important point, and mention that while their technique does indeed work using only 2 repeat cycle's worth of data, the identification of longer-term GZ change completely distinct from tidal variability (e.g. Friedl et al. 2019) may only be possible as/when more ICESat-2 observations become available.*

We have now included these two parameters in our automated workflow (line 187-189) and produced two figures (Figs. 7e and 7f) to show the number of repeat cycles and the mean tidal amplitude at each ICESat-2-derived GZ. We also discussed the uncertainties of GZ positions due to different ocean tidal amplitudes (line 297-300) and number of repeat cycles (line 329-336).

***Specific Comments***

***Line 6 –*** While it's beyond the scope of this review to provide detailed corrections to the style of English and grammar used throughout the manuscript, I've re-structured the abstract to greatly condense and more effectively convey the key selling points of paper here. The authors should feel free to amend/disregard as necessary. Authors should also note the omission of greater 'precision' relative to ICESat-1, as I strongly believe this is not convincingly addressed in the manuscript (see my comment above on line 190).

"*We present a new, fully automated method of mapping the Antarctic Ice Sheet's grounding zone using repeat-pass and cross-over analyses of newly acquired ICESat-2 laser altimeter data. Our method recovers the position of the landward limit of tidal flexure and the inshore limit of hydrostatic equilibrium, as demonstrated over the mountainous and hitherto difficult to survey grounding zone of Larsen C Ice Shelf. Since launching in 2018, our method has already doubled the number of grounding zone observations acquired from ICESat-1, which operated between 2003 and 2008. Acting as a reliable proxy for the grounding line, which cannot be directly imaged by satellites, our ICESat-2-*

*derived limit of tidal flexure locations agree well with independently constrained measurements, with a mean absolute separation and the standard deviation of 0.29 km and 0.31 km, respectively, from interferometric synthetic aperture radar-based observations. Our results demonstrate the efficiency, high spatial precision and density in which ICESat-2 can image complex grounding zones, and its clear potential for future pan-ice sheet grounding-zone mapping efforts."*

Thanks for the comment. We have now modified the abstract based on the reviewer's suggestions and removed 'precision'.

**Lines 18-21 –** I think these opening sentences could be worded more eloquently to say something like: *"Long-term satellite observations have linked the on-going thinning of Antarctica's ice shelves (Paolo et al., 2015) to enhanced rates of ice mass discharge beyond the grounding line (hereafter GL) – the point where the grounded ice sheet first detaches from the bedrock and begins to float (Fricker & Padman, 2006). Ice discharge calculations are sensitive ..."*.

Agree and amended.

**Line 28 –** Provide references for mass balance studies and those assessing overall ice sheet stability. The latest offering from the IMBIE crowd (Shepherd et al., *Nature*, 2018) would be good here.

Agree and amended.

**Line 33 –** Following my comment regarding Line 275, the authors should aim to avoid ambiguous terms like '*Its*' to begin sentences. Suggest rewording to *"The precise width of the grounding zone depends on ..."*.

Agree. This sentence has been modified and placed in line 338-339 "The width of the GZ depends on ice stiffness, bed slope and ice thickness (Bindschadler et al., 2011), and is useful in determining the ice thickness and rheology across the grounding zone (Dawson and Bamber, 2020)."

**Lines 35-36 –** This is a good example of many instances of repetition in the manuscript. The true GL (*G*) and limits of tidal flexure (*F*) have already been defined on line 30/31, so don't need to be repeated here. Saying something like the following would be much more concise and easier to understand.

*"... While Point G cannot be detected directly from satellite-based observations, Point F resides within close proximity of this location, and is thus widely considered to be the most robust satellite-observable proxy for the "true" grounding line (insert references here)."*

(Note that this rewrite also gets round the incorrect notion that the limit of tidal flexure is something only present on the ice sheet surface, when reality it is the surface manifestation of a variety of mechanical processes going on at the ice-bed-ocean interface)

Agree and amended (line 34-36).

I also suggest you replace any further mention of "*the true grounding line*", "*the landward limit*", "*limit of hydrostatic equilibrium*" etc. in the manuscript with 'Point *G*', 'Point *F*' and 'Point *H*'.

Agree and amended.

Also, in this instance '*Point*' is a pronoun, and so should always be capitalized. (be sure to change this universally throughout the text and fig. captions).

Agree and amended.

**Line 38** – Remove 'to date' (as it probably always will be ...).

Agree and amended.

**Line 41** – Change to "Points F and H (Figure 1) have previously also been derived using ICESat-1 laser altimetry ..."

Agree and amended (line 41).

**Line 43 –** The use of the word '*these*' is unneeded and can be removed.

Agree and amended.

**Line 44 –** Suggest replacing '*data*' with "*additional GZ information across regions* ...". Also hyphenate '*DInSAR-derived*'. For consistency with the above paragraph, please also refer to '*DInSAR-derived GZ information*' rather than '*GL information'* (since DInSAR recovers *F*, a \*proxy\* for *G*).

Agree and amended (line 43-44).

**Line 46 –** Change to "... *In addition, ICESat-1-based GZ detection is heavily based on visual interpretation which requires a large amount of manual* ..." for brevity.

Agree and amended (line 46-47).

**Line 51 –** Poor grammar. Suggest rewording and merging with the next sentence for clarity and conciseness. Something like: "*ICESat-2 has a repeat-pass orbit time of 91 days, and – compared to the single Geoscience Laser Altimeter System (GLAS) onboard ICESat-1 – measures the surface elevation of Earth's ice sheets using three photon-counting beam pairs emitted from the Advanced Topographic Laser Altimeter System (ATLAS)*".

Agree and amended (line 53-56).

**Line 58 –** Suggest shortening to: "... *Point H by analyzing repeat-track and ascending/descending cross-over data. We chose the* ...". The following sentence is also very long and verbose, and should be condensed and/or split up for clarity.

Agree and amended.

**Line 64 –** Saying that you used 12 months of data (5 cycles) when in reality the bulk of your analyses only used 8 (3 cycles) is somewhat misleading, so I'd consider splitting this paragraph up into 2 parts. One part discussing the repeat track analyses (8 months of data), and a standalone, follow-up part discussing the cross-over analysis (extended dataset covering 12 months). At present there's a lot of back-and-forth discussion between techniques, which took me several readings to fully understand what you meant.

Agree. We have now used the same period of data for both crossover analysis and repeat track analysis for consistency between the two methods (line 66-68) based on the comment by Reviewer 2.

***Line 69 –*** For absolute clarity, do these RPT's refer simply to the 'Pair Track' files included in the ATL06 products, or is this something you have calculated the position of? If the former, then I'd suggest just referring to them as 'pair tracks' (PTs) for consistency with ATL06 naming convention, and to avoid any possible confusion with the Reference Ground Track (RGT) information provided as part of the ICESat-2 orbit files (see: https://icesat- 2.gsfc.nasa.gov/science/specs).

The original statement is inaccurate, ICESat-2 points at the Reference Ground Track (RGT). However, we also noticed the similar statement has been made in Page 7 of the ATL06 ATBD document (https://nsidc.org/sites/nsidc.org/files/technical-references/ICESat2_ATL06_ATBD_r003.pdf): "Note that in cycles 1 and 2 of the mission, ICESat-2 did not point at the RPTs, and ICESat2's pairs are offset by up to 2 km from the RPT locations. The first cycle that was collected over the RPTs was the third", which is confusing in terms of RGT and RPT.

To avoid any ambiguity, we have now modified the sentence into "The first two cycles of ICESat-2 data acquired between 14 October 2018 and 30 March 2019 are not repeat cycles because the spacecraft pointing control was not yet optimized" in line 65-66.

***Line 83 –*** Should this say +/-15 m as opposed to 150 m? This value seems excessive, and I wonder how many values actually fell out with this range after your initial cleaning of the data?

We have removed this data filtering step according to the comment by Reviewer 2:

***Lines 81 – 82****: "In addition, the surface elevation was also compared with the reference DEM Reference Elevation Model of Antarctica (REMA) (Howat et al. 2019) provided in the ATL06 product..." This is extremely challenging in the region of the GZ. REMA elevations are not tide corrected; thus, their elevations in the GZ are time/scene dependent.*

***Lines 87-90 –*** Check grammar, punctuation and sentence tense. The first sentence is also repetitive and could easily be reworded to avoid double use of '*repeat track*'. Also, how/why does your cross-track slope correction 'facilitate' the automation of GZ mapping? Do you mean that it reduced the occurrence of incorrectly identified GZ features? I think a little more elaboration is needed here to explain this point. Sentence beginning line 90 should also be reworded to avoid the use of the word '*It*'.

Agree. We have now modified the sentence to "To identify the GZ features, we adopted a similar approach previously used with ICESat data by measuring the vertical motion of floating ice induced by ocean tides from a set of elevation anomalies for each repeat track (Brunt et al., 2010)". Separate discussions on the advantage of across-track slope correction are now in line 106-118 and line 213-218.

***Line 91 –*** Incorrect use of semi-colon and multiple unneeded uses of the work '*the*'. Also suggest changing '*features estimation*' to '*feature identification*'.

Agree and amended in line 89-91: "The workflow of the repeat track analysis developed in this study includes four steps: 1) Repeat-track data preprocessing (section 3.1.1, box 1 in Fig. 3). 2) Elevation anomaly calculation (section 3.1.2, box 2 in Fig. 3). 3) GZ feature identification (section 3.1.3, box 3 in Fig. 3). 4) Filtering and visual validation (section 3.1.4)."

***Paragraph beginning Line 93 –*** This is a good example of the 'back-and-forth' organization of the paper I mentioned in my general comments. That is, whereas reader expects this paragraph to involve a discussion of steps 1 and 2 detailed in line 91, the reader now has to go back to a discussion of repeat cycles and the grounding line reference product. In accordance with processing steps shown in Figure

3, I suggest the authors separate this part of the methods section into several distinct subheadings possibly labelled "*3.1.1: Repeat-track data preprocessing*" (to include discussion of nominal ref track calculation and search window size from reference GL dataset), "*3.1.2: Cross-track slope correction*" and "*3.1.3: GZ component identification*". If the authors choose to do this, then each section should also be explicitly cross-referenced in line 91. The processing steps corresponding to each sub-section could also be indicated on Fig. 3 by enclosing each sub-section in a box.

Good advice. We have restructured the methodology section into the following subheadings and modified Fig.3 by enclosing each sub-section in a box:
  i.    3.1.1 Repeat-track data preprocessing (box 1 in Fig. 3).
  ii.   3.1.2 Elevation anomaly calculation (box 2 in Fig. 3).
  iii.  3.1.3 GZ feature identification (box 3 in Fig. 3).
  iv.   3.1.4 Filtering and visual validation.

[Figure]

Figure 3. The ICESat-2 repeat-track workflow used to identify the grounding zone (GZ) in this study, including the limits of inland tidal flexure (Point F) and hydrostatic equilibrium (Point H).

***Lines 93-96 –*** Be sure to emphasize here that you used all three beam pairs in your analyses, as readers may interpret from the current version of the text that only one beam was used.

Agree and amended. We have now emphasized in the methodology section that we generated single beam repeat-track data groups for all six different ground tracks (line 98-105), and beam pair repeat-track data groups for all three beam pairs (line 106-113). Also see Figs. 4a and 4b.

***Line 99 –*** As per my comment on line 69. Should '*RPT*' be '*PT*'? Also consider putting inverted commas around '*nominal reference track*' to emphasize that this is a newly introduced term.

We have now removed 'RPT', and modified the statement to 'we also derived the locations of every reference segment for each ground track from the segment_quality group of the ATL06 product' (line 82-83) and 'For each single beam repeat-track data group, a 'nominal reference track' (black circles in Fig. 4a) at an along-track interval of 20 m was calculated by averaging the locations of reference segments from all repeat tracks. The use of reference segments to obtain the nominal reference track can produce a common set of geolocation profiles free of the data loss of actual ground tracks.' (line 102-105). We have put inverted commas around 'nominal reference track' in line 103.

***Line 101 –*** Needed to do what? I suspect set a GZ detection search window size, but this isn't obvious from the text.

We have changed the sentence into 'Before calculating the elevation anomalies for each track inside the repeat-track data group, a reference GL estimate is needed to define a GZ search window in the calculation (Fig. 3 box 2)' in line 120-121.

***Line 103 –*** Why was the Mouginot dataset used over the ice rise information documented in Depoorter? Is it more accurate? More up-to-date? Explicitly state why you used this product here.

We have removed the Mouginot et al. (2017) grounding line product from the manuscript and used the Depoorter et al. (2013) grounding line for both the ice sheet and ice rises or ice rumples. As mentioned in the previous responses, we have now modified the reference GL by merging the latest ESA CCI DInSAR GL (2015-2016) with the Depoorter et al. (2013) GL to avoid the uncertainty of the GL location in the calculation (line 120-129).

***Line 104 –*** What is an '*initial GL product*'? I see no mention of it in the text or Figure 3, and don't know how it differs from the 'reference GL product' discussed in the same sentence.

The 'initial GL product' is the reference GL used in the algorithm. We have changed the sentence into 'For each repeat-track data group, the reference GL was calculated as the intersection between the nominal reference track and the composite GL.' in line 130-131 for clarity.

***Line 111 –*** In 2017 Larsen C calved massive iceberg A-68 from its ice front, exposing an ~6000 km2 area of previously ice-shelf covered ocean water. Mouginot's coastline precedes this calving (and the ICESat-2 observational period), meaning that the authors have inadvertently included this huge section of open water in their analyses. This is clearly seen in Figure 2. I suggest the authors re-clip their dataset using a more up-to-date coastline, such as the one available at: https://www.add.scar.org/. The authors may also wish to consider clipping the velocity dataset shown in Figure 5 for consistency between figures.

Agree. We now use the SCAR Antarctic Digital Database (ADD) coastline to mask out the ATL06 data located in open water, please refer to line 136-138. All the related figures (Figs 2, 7, 10 and A1) have also been modified by using the ADD coastline mask.

**Line 112 –** This sentence is highly repetitive and includes '*elevation change*' three times. Consider condensing.

We have now removed this sentence.

**Line 119 –** Insert comma after $h_{Li,G}$. Also, I presume here that '*Li*' means linearly interpolated? Your choice of interpolation isn't explicitly stated here, so if so, include that info here.

'Li' means land ice, for clarity we have now removed all the use of 'Li' in the equations (line 145-152).

**Line 125 –** "... *are the y coordinates measured perpendicular to the RGT*". Can this be reworded to say "... *are the across-track y-coordinates measured perpendicular to the RGT*"? (I guess, after all, that this is what '*ATC*' in e.g. '$y_{ATC, x}$' means?)

The reviewer is correct, we have amended the sentences in line 147-148 and line 152.

Also, following my comments on Line 69, the use of *RGT* here particularly qualifies the need to abbreviate '*Reference Pair Track (RPT)*' to '*PT*' in the text.

The use of RGT is correct here, please refer to the definition of y_atc provided in the ATL06 variable dictionary (https://nsidc.org/sites/nsidc.org/files/technical-references/ICESat2_ATL06_data_dict_v003.pdf): 'Along-track y coordinate of the segment, relative to the RGT, measured along the perpendicular to the RGT, positive to the right of the RGT'.

**Line 127 –** Suggest changing "*in GZ estimation*" to "*used in our identification of the grounding zone*". Remove unneeded period at the end of the last sentence.

Agree and amended in line 154.

**Line 129 –** Suggest changing to "*After applying the cross-slope track...*". Also consider splitting this very long sentence in two between '*track*' and '*which*', and opening the new sentence with "*These profiles are shown in Figure 5a*".

Agree and amended in line 154-156.

Also note here the irregular ordering of the Figures in text. Figures should generally be referenced in the text in alphabetical order, so in this instance I suggest swapping Figures 5a and b in Figure 5, and changing Line 131 to say '*5a*'. Same for line 132 ('*5b*'). Also, remove "*color-coded dot curves*" here and in line 132; this is non-essential information best left for the caption.

Agree and amended.

**Line 133 –** Suggest: "*To estimate Points F and H from the elevation anomalies ... was calculated from the corrected ...*"

Agree and amended in line 162-163.

**Lines 136-160 –** Following my general comment, these paragraphs are very difficult to follow, as a) they flick back and forth between the identification of *F* and *H*, and b) it is highly unclear what 'new' identification procedures the authors have used versus that discussed in Fricker & Padman (2006). There is also a lot of repetition of content already mentioned in the introduction. I strongly suggest re-structuring these paragraphs to closely follow the structure outlined below:

First paragraph: Very brief overview of the anatomy of the GZ with respect to Points *F* and *H* and the work of Fricker & Padman (1-2 sentences). Then, a few sentences on how *they* identified the grounding zone.

Second paragraph: A discussion of how the authors expanded upon this earlier work and identified Point *F* using new slope-corrected techniques, MAEA and its second derivative. No mention of *H* here.

Third Paragraph: Discussion of how using MAEA alone is likely more difficult to identify Point *H*, and subsequently: how the authors used MAEA together with their error function to locate *H*. When discussing the error function, elaborate more on what "*as a guide*" means (readers should be able to replicate this method, so as written this seems vague/ambiguous).

Agree and we have modified the structure of the Section 3.1.3 'GZ feature identification' based on the reviewer's suggestion.

**Line 163 –** unneeded '*the*' before elevation change. Also, this and next sentence are almost complete repetition, and should be condensed (and possibly merged with the following sentence). Also define '*KDTree*' (and any other abbreviation) in full when using for the first time.

Agree and amended.

**Sentence beginning line 168 –** Poor grammatical structure.

Agree and amended.

**Line 170 –** Should this say '*from*' instead of '*for*'?

Agree and amended.

**Line 175 –** Add Padman et al. citation after reference to the CATS2008a model. '*Movement*' should be singular.

Agree and amended.

**Line 178 –** '*based on the results in section 4*'. What does this mean? State the explicit steps you took to derive this number here, and then refer to the results section, if still necessary.

The threshold for removing crossovers with the same tidal phase on ascending and descending track is defined based on the minimum tidal amplitude detected from the repeat track analysis, which can only be achieved by analysing all the calculated GZ results. We have now changed the statement into "As the minimum detectable tidal amplitude from repeat track analysis is 20 cm after analysing all the GZ features calculated in Section 3.1, we then set the minimum threshold of elevation change due to ocean tides on floating ice measured by the crossover analysis to be 40 cm by doubling the 20 cm threshold of repeat track analysis." In line 240-243.

***Sentence beginning line 180** –* It is implicit in the above sentence that you did this, so suggest the removal of the sentence from the text.

We think it is necessary to keep this sentence. We only mentioned that the tidal amplitudes calculated from the CATS2008 Tidal Model were used as a reference for the vertical motion of floating ice but we didn't mention how we were going to use CATS2008 Tidal Model tidal amplitudes to filter the crossovers. This sentence provides important information on the method of filtering crossovers with the ascending and descending track in the same tidal phase.

***Line 183** –* suggest rewording to better match the structure of my suggested abstract changes. Suggest something like:

"*Using our newly developed ICESat-2-based GZ detection algorithm, we recover a two-fold increase in the number of GZ features identified over Larsen C Ice Shelf by ICESat-1 (69 and 71 picks of Points F and H, respectively, versus 30 of each by ICESat-1). The spatial distribution of each platform's GZ picks is shown in Figure 6, together with a comparison of Point F as determined from independent DInSAR observations. The improvement in our ability to image the GZ using ICESat-2 data is especially notable in heavily crevassed regions like Jason Peninsula and Churchill Peninsula (Jansen et al. 2010), which were previously difficult to image using single-beam ICESat-1 observations alone*".

Agree and amended in line 249-260.

***Lines 189-195** –* As per my technical comments on Lines 183-216. Suggest restructuring this paragraph and Figure 5 to only make reference the more up-to-date ESA CCI product.

Agree and amended in line 276-288.

***Lines 189-201** –* This paragraph is another example of the somewhat unorganized writing style discussed above, this time pertaining to the back-and-forth comparison of different sensor observations. Mirroring my suggested edits to the methods section regarding the derivation of *F* and *H*, I suggest re-structuring this and the following paragraph in the following order to greatly improve clarity.

First para: Discussion of how the author's results compare with DInSAR observations (ESA CCI only). (Any comparison of ICESat-1 and -2 GZ points are largely unconvincing and should be excised from the text here).

Second para: Discussion of where your DInSAR and ICESat-2 *F* points don't agree (i.e. the para beginning line 202. Nice findings, by the way!).

Third para: Discussion of how your *F* points compare with those imaged by other sensors, including CryoSat-2 (nb. A discussion of CryoSat-2 vs. ICESat-1/DinSAR observations is outwith the confines of this study, and as a predominantly ICESat-2 methods study, should be removed. Also note the incorrect spelling of *CryoSat-2*). In accordance with my earlier comments, I think this paragraph would also be the place to add in comparison of your $I_b$ picks vs. those of the ASAID product).

Fourth Para: Discussion of Point *H*, encompassing my comments on lines 233-235 above.

Agree and amended in Section 4.2 'Comparison with other GZ products'

**Lines 206-210** – *"(vertical red dash line)"*, *"(blue dash curve)"* etc. is non-essential information best placed in the caption instead.

Agree and amended.

**Lines 209-210** – Highly verbose sentence, and repetitive use of 'left' and 'right'. This sentence should be condensed.

Agree and amended.

**Line 220** – Insert comma before and after $|dh|$.

Agree and amended.

**Line 221** – Change "*the repeat track analysis*" to "*our repeat track analysis*"

Agree and amended in line 308.

**Line 223** – What does "*same time difference*" mean? Suggest rewording for clarity.

We have removed this example from the manuscript.

**Line 224** – Suggest changing to "*The locations of Point F identified using our repeat-pass technique (black dots in Figure 8c) are located in the middle of high and low $|dh|$ observations, indicating the presence of floating and grounded ice either side of Point F, respectively*".

We have removed this example from the manuscript.

**Paragraph beginning line 227** – For simplicity and clarity, I would restructure this paragraph to first mention why we're interested in knowing the width of the GZ, then discuss how it has been calculated in the past and then how the authors have calculated it in this paper. Then finish with a discussion of your results. Similar to the discussion of Points *F* and *H* within the context of independent observations, some brief words on how measured GZ widths compare with e.g. ASAID GZ widths would also be well placed here. Also, the first sentence of this para is repeated directly from the introduction, so suggest rewording.

Agree. We have restructured and modified Section 4.3 'Grounding zone width'. Since ASAID GZ was mapped between Points I and H but our method is based on the separation between Points F and H, we think it is inappropriate to compare our GZ width with ASAID GZ width. Instead, we compared our results with the most recent CryoSat-2-derived GZ width on Larsen C Ice Shelf by Dawson and Bamber (2020), they measured Points F and H based on a similar method in detecting tidal flexure information.

**Line 239** – Suggest changing to: "*... for recovering several commonly imaged components of the ice-sheet-shelf grounding zone, including the inland limit of tidal flexure (Point F) and the limit of hydrostatic equilibrium (Point H). The new method presented in this study should ...*"

We have removed this sentence from the manuscript.

**Line 240** – "*The new automated method presented in this study should allow a more efficient and consistent mapping of the Antarctic GZ features*". I see what the authors are trying to suggest here, but don't think it's articulated quite correctly. The authors should bear in mind that due to recent

advances in polar-orbiting EO capabilities (incl. the launch of Sentinel-1a/b), we can in theory now generate a precise understanding of the GZ every 6 days using DInSAR. The spatial-temporal coverage of e.g. DInSAR-derived *F* will therefore *always* be better than ICESat-2 *F*, so I think this sentence (and any similar arguments made further down in the discussion and conclusion sections) should be re-worded to reflect this. What I think the authors want to say is that ICESat-2 provides an automated, efficient means of characterizing the Antarctic Ice Sheet's GZ, to complement the high-accuracy observations afforded by other modern Earth Observation sensors.

We have removed this sentence from the manuscript.

*Line 241 –* This sentence should go into more detail about why we are interested in GZ position change and width (e.g. for improving numerical models aiming to constrain SLR predictions, identifying Antarctica's most vulnerable regions to climate change etc.).

We have removed this sentence from the manuscript. The reasons why we are interested in GZ position change are in Section 1 'Introduction' (19-29).

*Line 244 –* Suggest rewording to "*To maximize repeat-track coverage over the GZ, our method considers the ground tracks of one ICESat-2 repeat cycle beam pair as two individual repeat tracks*".

We have removed this sentence because we have now changed the ATL06 data used in this study and the repeat track analysis method, please refer to Sections 2 and 3.

*Line 247 –* Should say "*... which was an issue when using ICESat-1 data to identify the GZ (Brunt et al, 2010) ...*". *Estimation* implies imprecision, so *identify* or similar is better".

This sentence has been removed.

*Line 248 –* "*In addition, this also allows us to calculate the GZ features from just two time stamps of tidal amplitudes*". It's obvious to me why the authors say this, but to some this might imply an element of inaccuracy in the author's results. This is because recent research has shown that the GL can undergo significant changes in its position over a range of tidal frequencies, in some cases by as much as 4 km (e.g. Milillo et al., 2017, *GRL*).

I suggest the authors acknowledge this important point, and mention that while their technique does indeed work using only 2 repeat cycle's worth of data, the identification of longer-term GZ change completely distinct from tidal variability (e.g. Friedl et al. 2019) may only be possible as/when more ICESat-2 observations become available.

Agree and amended in line 329-336, please also see our previous response to the reviewer's technical comment on line 277.

*Line 240 –* see my comments re: Line 240. While this was perhaps once true, with the launch of Sentinel-1a/b (and soon -1c) I think the authors should reword this to state how the two datasets will *complement* each other.

We have removed this sentence from the manuscript.

*Paragraph beginning Line 257 –* This paragraph is a bit 'matter-of-fact' or bullet-point-like, and at times not merited within the context of this paper (e.g. the review-like discussion of CS2 vs. ICESat-1 observations, which are independent of the method presented here). In any case, for simplicity I

would reword the paragraph to echo the structure the results section (see my comments on Lines 189-201), and again excise any discussion of ICESat-2 vs. ICESat-1 precision from the text. As this is the discussion section, a regurgitation of the exact separation distances is also not essential here. Also, to avoid any possible confusion, I would refrain from referring to '*GL detection*' when referring to non-Point *G* components of the GZ.

Agree and amended in Section 4.2.

***Line 264*** – " *ice plains that are difficult to capture from DInSAR* ...". Careful! Sentinel-1 DInSAR can capture it just fine (see below), so I'd reword this.

[Figure]

**Figure 1** – Churchill Peninsula ice plain detected from double-difference InSAR applied to 2019 Sentinel-1a/b data.

Agree, this statement has now been removed.

***Line 268*** – "*..., it demonstrates*". What demonstrates?

We have now changed this sentence into "In addition, proving our method works at a 20 cm scale tidal amplitude in a region with complex relief demonstrates the ability for the generation of GLs for the majority of the Antarctic Ice sheet" in line 300-312.

***Lines 270/271*** – Where are these tidal estimates derived from? CATS2008a? Cite the appropriate reference here. Also, remove the space separating this and the next paragraph.

They are from CATS2008 Tidal Model, we have now added the citation in line 304.

***Lines 279*** – Suggest rewording to something more concise like "*Surface elevation changes derived from ICESat-2 cross-over data also provide valuable information about the approximate location of Antarctic grounding zones*".

Agree and amended in line 314-315.

**Line 281 –** should say "... *separation afforded by repeat track observations, it can still reveal the approximate location of the GZ*". I also suggest the next sentence reads: "*In doing so, this method has the potential to provide important validation of repeat-track-derived GZ features, including along fast-flowing ice streams where the GZ often undergoes rapid changes ...*" or similar.

Agree and amended in line 315-317.

**Line 283 –** Restructure sentence to condense and avoid the use of '*Also*...'. This and the next sentence could also be easily merged.

Agree and amended in line 317.

**Line 288 –** This sentence is rather verbose, and I think it could easily be shortened and/or restructured to convey the key selling point of paper more effectively (see my wording of the abstract above).

Agree and amended.

**Sentence beginning Line 288 –** '*the* 8-month period'.

Agree and amended, now is the 10-month period of data.

**Sentence beginning Line 293 –** Larsen C Ice Shelf is a pronoun and so the preceding use of the word '*the*' is incorrect and should be removed. Next sentence once again begins with '*it*'. What does '*it*' refer to? Reword for clarity.

Agree and amended.

**Sentences beginning Line 294 –** I would let the readers decide if the separation between your ICESat-2 and DInSAR observation are small, so remove '*small*' here. Also change all instances of '*GL*' to '*GZ*' for accuracy, and hyphenate all uses of the word "*derived*".

Agree and amended.

**Line 296 –** Again, I'm just not sure you can confidently talk about ICESat-2 vs. ICESat-1 precision that way. Also, as far as I can tell, in the current version of the manuscript the authors make no direct comparison between their ICESat-2- and any CryoSat-2-derived *F* products, so don't think any robust statements can be made regarding precision in that sense either. As a result, I think this sentence should be restructured to echo my suggested restructuring of the results section.

Agree and we have removed this sentence.

**Line 297 –** Change '*this method'* to '*our method'* and remove '*the*' before Larsen C Ice Shelf.

Agree and amended in line 364.

**Line 298 –** Insert a comma after '*range*'.

Agree and amended in line 365.

**Line 299 –** rephrase to avoid "*it still shows*". Also '*GL'* should say '*GZ*'.

Agree and amended.

**Line 301 –** Suggest changing to *"...Antarctic Ice Sheet, similar cross-over-based analyses have the potential to provide more accurate depictions of the GZ"*.

Agree and amended in line 370-371.

**Sentence beginning line 301 –** This is almost complete repetition of lines 293-295, so I suggest removing it from the text.

Agree and removed.

**Line 310** – Where can the ESA CCI DInSAR data be found?

We have now added 'We thank the European Space Agency (ESA) for providing the ESA CCI DInSAR grounding line data' in line 378-379.

**Line 333 –** Incorrect dataset cited.

Agree and amended.

**Figure 1** – It may just be the way the PDF has rendered the image, but as printed, the vertical lines pointing to each of the GZ components have different line thicknesses and slightly different colors. Suggest the authors standardize all thicknesses and colors if this is not a rendering issue.

We have now standardized all the thickness and colors of the plot.

**Figure 1 caption** – Capitalize all instances of '*Point*'.

Agree and amended.

**Figure 2 –** As previously discussed, suggest the authors clip their dataset to the modern-day coastline (i.e., account for A-68's calving where elevation clearly equals zero over open water in the fig) and only show the 2017 (ESA CCI) grounding line. I also wonder if the authors inadvertently clipped Bawden Ice Rise out of their analyses? Like Gipps Ice Rise, this is an important pinning point for Larsen C (c.f. Borstad et al., 2017; *GRL*), so it'd be interesting to see any available GZ picks here. I suggest also standardizing the font size of all labels on the main map, and instead assigning track labels a different font color to better contrast against place-names. I also suggest center justifying the scale bar relative to the other items of the legend, and moving all items slightly to the left (at the moment the 1000 m label intercepts the figure border and looks untidy). Taking potentially color-blind readers into consideration, I'd also encourage the authors to choose another, non-jet/rainbow color ramp.

We have now clipped the datasets using the ADD coastline mask, changed font size and colors of the labels, adjusted the legend and changed the colorbar. We used the Depoorter et al. (2013) GL in the background to show the complete GL coverage in this region especially for Bawden and Gipps Ice Rise and Hearst Island where ESA CCI grounding line is not available.

**Figure 2 caption** – Capitalize '*Ice Shelf*' and amend caption to reflect the addition of the 2017 (ESA CCI) grounding line. Remove '*in Antarctica*' as the inset clearly shows this. As the grid is shown in meters, please also state the coordinate reference system used (e.g. EPSG:3031 Antarctic Polar Stereographic) (same for Figs. 6 and 8).

*Agree and amended.*

**Figure 3** – As per my comments on the paragraph beginning Line 93, authors might consider enclosing the various sub-sections of their methodology with boxes. '*Nominal Reference Track*' and '*Hydrostatic Point H*' should also be shifted so as to not impinge on the boxes. Capitalize '*Point F*' and '*Point H*'.

*Agree and amended.*

**Figure 3 caption** – Suggest "*The ICESat-2 repeat-track workflow used to identify the grounding zone (GZ) in this study, including the limits of inland tidal flexure (Point F) and hydrostatic equilibrium (Point H)*" (note: In general, abbreviations should be written in full in figure captions, which is why I have done so here).

*Agree and amended.*

**Figure 4 caption** – Suggest either amending the figure or stressing in the caption that this image only refers to 1 of the 3 beam pairs imaged by ICESat-2 at any one time (not all readers will be familiar with the 3-beam imaging capabilities of ICESat-2).

*Agree and amended.*

**Figure 5** – Why is repeat cycle 5 not shown? In the main text it is stated that this dataset is used, so its omission here is rather odd. Suggest also ordering the repeat cycle legend in numerical order (*3l, 3r, 4l, 4r* and so on). Also, I don't quite understand the inclusion of elevation by latitude (dashed profiles) here? The only reason I can think of is because track 1192 overpasses two GZs on the north and south of the unnamed ice rise seen in Fig.2 (and so should have some easily identifiable break-in-slope), but even then it's unclear where this happens. In any case, when superimposed over the anomalies and other info it makes for a rather busy diagram, so I'd suggest removing it for simplicity. If the authors feel this is essential information, then I recommend they include it either as an additional sub-panel to this figure (cf. Fricker et al., 2006) or as a supplement.

In all panels, some of the axis and other labels appear cut-off, so care should be taken to fix these minor blips.

*We have now included all the repeat cycles (3, 4, 5 and 6) used in this study and reordered the repeat cycle legend in numerical order. We changed the figure into a similar layout as Fig.3 in Fricker and Padman (2006) to improve the overall clarity.*

**Figure 5 caption** – This is a very long and difficult to comprehend sentence (and caption overall). Can it be condensed somehow? I'm also not sure the second sentence makes grammatical sense, and could be shortened to: "*in the legend, c03_1r refers to the right ground track of ICESat-2 pair 1, repeat pass 3*". After the Padman reference, insert "*following Fricker & Padman, 2006*". Information about the derivation of MAEA has already been discussed in the text, and so should not be included in the caption. Capitalize all instances of the word '*Point*'.

*Agree and amended.*

**Figure 6** – Same comments as Figure 2 regarding the center justification and movement of the items in each of the legends. Authors may also wish to consider clipping the velocity map for consistency with Fig. 2. Update DInSAR coverage to only show the 2017 Point *F*. In *c* what do boxes *A* and *B* refer to? (This isn't stated in the caption). I think they refer to the spatial extent of Figs. 8a/b - if so, suggest

labelling '*Fig. 8a*' and '*Fig. 8b*' here, and similarly indicating the extent of Fig. 8c. Also standardize the thickness of the DInSAR Point *F* line in all plots.

Agree and amended, this figure is now Fig. 7.

***Figure 6 caption*** – Once again, '*estimates*' inherently suggests imprecision, so suggest rewriting caption to begin: "*ICESat-2-derived inland limits of tidal flexure (Point F; red dots). For comparison, ICESat-1-derived Point F (Brunt et al., 2010) picks are also shown (blue dots), and data are superimposed over on recent ice surface velocity magnitudes (Rignot et al., 2017). b) same as a), but showing ...*". Then, at the end, "*in all plots, DInSAR derived Point F is also shown*".

Agree and amended.

***Figure 7*** – As per my comments on Fig 5. Also, is there any way to re-size the labels in *c* and *d* so that they don't cross over the reference lines and data? At present these look messy.

Agree and amended, this figure is now Fig. 8.

***Figure 7 caption*** – Another long and verbose caption which, being very similar to the caption of Figure 5, can be easily shortened to "*Same as Figure 5, but showing ...*".

Agree and amended.

***Figure 8*** – In all subplots, show only the 2017 ESA CCI Point *F* lines. In *c*, a black circle appears to be obscuring a blue circle. Suggest using unfilled black circles (i.e. outline only) in all subplots to avoid this. Suggest also scaling the black circle in the legend to match the its size as shown in the sub plots.

Agree and amended, this figure is now Fig.9. We have now changed this figure to only include the crossovers on Hearst Island, we kept using the filled black circles because the current figure doesn't have the same overlaying issues as the previous submission.

***Figure 8 caption*** – Incorrect grammar. Suggest "*Spatial distribution of ICESat-2 crossovers analyzed in this study. The spatial extent of plots a-c is shown in Figure 6c. At each location, the absolute change in elevation, |dh|, is shown. Background is ...*".

Agree and amended.

**References cited in this review**

Adusumilli et al., 2017, *Geophysical Research Letters*, doi:10.1002/2017GL076652 Bindschadler et al., 2011, *The Cryosphere*, https://doi.org/10.5194/tc-5-569-2011

Borstad , McGrath and Pope, 2017, *Geophysical Research Letters, https://doi.org/10.1002/2017GL072648*

Fricker et al., 2012, *Journal of Geophysical Research: Oceans*, doi:10.1029/2011JC007126, 2012

Scambos et al., 2007, *Remote Sensing of Environment*, https://doi.org/10.1016/j.rse.2006.12.020

Shepherd & the IMBIE team, 2018, *Nature*, https://doi.org/10.1038/s41586-018-0179-y

**Reviewer 2**

**Mapping the Antarctic Grounding Zone from ICESat-2 Laser Altimetry Tian Li, Geoffrey J. Dawson, Stephen J. Chuter, Jonathan L. Bamber**

This paper uses ICESat-2 data to identify the grounding zone of the Larsen C Ice Shelf, Antarctica. It builds off of methods developed for ICESat and automates some of these processes. The authors describe a modified repeat-track analysis (that uses 6.5 months of ICESat-2 data) and a crossover analysis (that uses 1 year of data) to determine a GZ that is in close proximity to the continuous GL assessment from the DInSAR GL of Rignot et al., 2016; this agreement is heavily based on the increased resolution of the ICESat-2 data, and not a measure of precision, as the authors contest.

The paper overall needs a fair bit of rewriting, with some text being repetitive and other sections feeling out of place. More care needs to be taken with respect to using uniform language with respect to terms that are introduced. And more care needs to be taken with respect to references ('et al.' is often, but not always, missing).

The method presented here builds off of methods developed for ICESat (please change all references of 'ICESat-1' to 'ICESat'). Some of the modifications may be relevant (such as the across-track slope correction) but are not strongly justified; other changes (or additions) do not seem to add to the results on this ice shelf (such as the crossover analysis).

Thanks for the comments, please see the major changes we have made in light of the comments by both the reviewers in our response to Reviewer 1.

***My major comments include:***

1) Why do the authors utilize a across-track slope correction and migrate the analysis to the 'nominal reference track'? In line 97, the authors state that the ICESat-2 "separation between same side ground tracks in a pair from different repeat cycles is less than 10 m". I do not expect substantial across-track slope, or rough terrain, at 10-m length scales. Uncorrected elevations in Figures 5a and 5b confirm this. The authors could instead produce repeat-track assessments through the GZ for all 6 ground tracks, increasing the number of point F and point H assessments substantially. If the across-track slope correction is needed for more subtle reasons, the authors need to better justify this.

This is a very good point. At the time when we developed this method, there wasn't a sufficient number of repeat cycles, so we treated the two beams in a beam pair as two individual 'repeat tracks' to maximise the data we could use, as using a small number of repeat cycles (i.e. 2) on individual beams resulted in noisier data especially in high-sloping regions. With the addition of more cycles, we have now been able to include the repeat track analysis for individual beams (Section 3) in our study. However, in some high-sloping areas such as the case for the left beam in track 506 beam pair 2 (Figs. 6a-6d), it is clear that a ~10 m separation cannot solve large across-track slope and resulted in a wrong GZ identification, but the across-track slope correction can greatly reduce this error (Figs. 6e-6h). In addition, the high elevation anomalies around the coupling point shown in Figs. 8b and 8c also show that the across-track slope can be significant over a 10-m scale.

Therefore, we think it is important to keep the across-track slope correction for beam pair in our methodology for high-sloping regions, at the same time we also performed the analysis for single beam repeat tracks to increase the density of GZ observations. The modified methodology is in Section 3, the discussion of the GZ identification by these two methods is in line 261-272.

2) I do not believe that the crossover analysis provides any additional merit to the existing repeat-track analysis for the Larsen C. As the authors mention, it might be more useful in more southerly ice shelves. I suggest removing that element from the paper and using it at a later date.

We have now used the latest version 3 ATL06 data for crossover analysis. We also added an additional step - 'For crossovers calculated from different cycles at the same location, we also averaged the elevation differences at each crossover location' (line 245-246), this removes a lot of noise observed in the previous data. The distribution of crossovers |dh| on Hearst Island (Fig. 9) shows a clear transition from land ice (low |dh|) to floating ice (high |dh|), which aligns well with repeat-track-derived Point F and the REMA break-in-slope. Although the spatial separation between crossovers is much larger than the 20 m along-track separation afforded by repeat track analysis, this example still demonstrates how crossover analysis can be used to validate the repeat-track-derived GZ feature, and shows the potential for the application of this method to southern regions of the Antarctic Ice Sheet.

3) The authors state that the agreement between this GZ and the GL of DInSAR is an indicator of increased precision, relative to ICESat; 'precision' is not a great term for this. The close agreement is undoubtedly due to the increased across-track and along-track data density associated with ICESat-2; this is a spatial resolution improvement, not a 'precision' improvement.

We agree that the close agreement between ICESat-2-derived GZ and ESA CCI DInSAR GZ observations are benefited from the increased across-track and along-track density of ICESat-2 data. We have now removed the word 'precision'.

**Specific comments:**

**Throughout:** 'ICESat', not 'ICESat-1'

Agree and amended.

**Throughout:** Clean up the references (missing 'et al.' often)

Agree and amended.

**Title:** this is a bit misleading, suggesting that this is a result for the entire Antarctic. The definite article is not appropriate here. I suggest somehow adding 'Larsen C' to the title.

Agree and we have changed the title into 'Mapping the Grounding Zone of Larsen C Ice Shelf, Antarctica, from ICESat-2 Laser Altimetry'.

**Line 10:** the results *should* show a three-fold increase; why only 2?

Using the modified repeat track analysis method in Section 3, our results now show a near nine-fold increase in ICESat-2-derived GZ features compared with ICESat, this agrees well with the nine sets of repeat-track data groups generated in our repeat track analysis workflow.

**Line 30:** Point I is poorly characterized in Fig 1 and not mentioned in this paragraph. Suggest removing it from figure.

We have now modified Fig. 1 to better characterize Point I and included the relevant explanation on Point I in line 33 and line 48-50.

***Lines 49 – 51:*** "The launch of ICESat-2 on 15 September 2018, as a successor to the ICESat-1 satellite mission, can achieve higher along track spatial resolution as well as nine times better coverage compared with its predecessor, providing the potential to map the GZ with greater accuracy and spatio-temporal coverage." This needs to be better described. There are both along-track and across-track data density issues here. And there are more repeats within a calendar year. But that's not captured explicitly in this text.

Agree and amended in line 65-72.

***Lines 64 – 68:*** The time span is unnecessarily complicated in this section. A couple options: 1) remove the crossover analysis. 2) reduce the time span of the crossover analysis to match the repeat-track analysis.

Agree and we have now used the same period of data for both crossover and repeat track analyses (line 65-72).

***Lines 67 – 68:*** Note that ICESat-2 was not operational for much of July. And the analysis stops halfway through November. Thus, there are really only 6.5 months of repeat track analysis, and cycle 4 (?) is not a complete cycle.

Agree and amended in line 66-70.

***Line 69:*** "... ICESat-2 did not point at the Reference Pair Track (RPT)" Careful here; ICESat-2 points at the RGT; the RPTs are specific to the ATL06 data product.

We agree with the reviewer that ICESat-2 points at RGT based on the definition of RGT from the ATL06 ATBD (https://nsidc.org/sites/nsidc.org/files/technical-references/ICESat2_ATL06_ATBD_r003.pdf): 'The reference ground track (RGT) is the track on the earth at which a specified unit vector within the observatory is pointed'. However, we have noticed the similar statement on RPT was made in Page 7 of the ATBD "Note that in cycles 1 and 2 of the mission, ICESat-2 did not point at the RPTs, and ICESat2's pairs are offset by up to 2 km from the RPT locations. The first cycle that was collected over the RPTs was the third". To avoid any ambiguity, we have now modified the sentence into "The first two cycles of ICESat-2 data acquired between 14 October 2018 and 30 March 2019 are not repeat cycles because the spacecraft pointing control was not yet optimized" in line 65-66.

***Line 73:*** The start of this paragraph could be smoother; something like, ICESat data were corrected for ocean tides and had to be retided; ICESat-2 ATL06 data are not...

Agree and amended in line 74-76.

***Lines 81 – 82:*** "In addition, the surface elevation was also compared with the reference DEM Reference Elevation Model of Antarctica (REMA) (Howat et al. 2019) provided in the ATL06 product..." This is extremely challenging in the region of the GZ. REMA elevations are not tide corrected; thus, their elevations in the GZ are time/scene dependent.

Agree and we have now removed this data filtering step.

***Line 83:*** "elevation differences with REMA fell outside a ±150 m threshold" What? 150 m of elevation differences?

We have now removed this data filtering step.

*Line 108:* "300 m were removed as the surface elevation of GZ" How many points did this remove? Were many of these points picked up by the ATL06 quality summary?

We set the 300 m threshold to remove ATL06 elevations located on land ice that are not needed in the GZ calculation, this improved speed of the analysis (line 134-136). As this is not a quality filtering for the ATL06 data, it is inappropriate to compare with the ATL06 quality summary flag.

*Line 112:* "The elevation change calculated from repeat tracks is a combination of cross-track slope induced elevation change ..."You say above that the repeatability is 10 m; I do not think that there is appreciable slope on 10-m length scales. I think what you are trying to say is that 1) you are prescribing height changes to the Nominal Reference Track (language from figure) or the RPT (language from above); 2) slope on 90 m length scales are important. Suggest 2 things here: 1) get the language between figure 4 and the text above more common; and 2) state from the onset that your analysis is being done on the RPT. Although, as I stated above, I do not think a slope correction is needed with 10 m repeatability.

Please refer to our response to the first major comment. We have now explicitly explained in the manuscript that the calculation was performed along the nominal reference track for each repeat-track data group, which is the average of all reference segments of each track (Section 3.1.1).

*Lines 113 – 114:* "As only 8 months of repeat cycles from March to November 2019 were used, ..." As stated above, it's really only 6.5 months with the loss of July.

Agree, the current study period is actually 10 months.

*Line 121:* Define these terms before equation 2 is introduced. Spell out h_LI is (I assume) height for 'land ice'; but what is y_ATC? What does ATC stand for?

Agree and amended, please see our response to Reviewer 1.

*Line 125:* RGT? Or the actual surveyed ground track?

The use of RGT is correct here, please refer to the definition of y_atc provided in the ATL06 variable dictionary (https://nsidc.org/sites/nsidc.org/files/technical-references/ICESat2_ATL06_data_dict_v003.pdf): 'Along-track y coordinate of the segment, relative to the RGT, measured along the perpendicular to the RGT, positive to the right of the RGT'. Please also see our response to the comment by Reviewer 1.

*Line 126:* Nominal? Or calculated mid-point for the actual surveyed ground tracks?

Here we use the nominal reference track, which was calculated by averaging all the reference segments for each ground track stored in 'segment_quality' group of ATL06 data (line 82-84 and 102-104). The reason of using reference segments to calculate nominal reference track is to produce a common set of geolocation profiles free of the data loss of actual surveyed ground tracks.

*Line 126:* "The average of all cross-track slope corrected elevations..." I'd be clearer and uniform with language. For clarity, start with 'elevations' (that's what they are) and then use a single term throughout (e.g., the start of the next paragraph); I'd call these 'elevations corrected for cross-track slope'.

Agree and amended.

*Line 129:* "After the cross-track slope correction for each ground track in a pair, ..." This is not clear.

Agree and amended in line 154-156.

*Line 130:* 'reference elevation profile ...' Use uniform language when defining terms; make this match the language from the last paragraph.

Agree and amended.

*Lines 131–132:* Watch the order of how you introduce Figures.

Agree and amended.

*Line 141:* "consistent with the local tidal amplitude..." Cite the source of that amplitude.

Agree and amended.

*Line 152:* This paragraph needs a rewrite; it's overall confusing

Agree and amended (Section 3.1.3), please see our response to Reviewer 1.

*Line 158:* "predictions at 10 km offshore..." 10 km may be too large to extend the methods to complex GZs; suggest 1 km.

We have changed this value to 5 km (line 184-186). We don't think 1 km is an appropriate choice here as the average GZ width is typically around 5 km, ~3.2 ± 2.6 km for Ross Ice Shelf (Brunt et al., 2010) and ~5.2 ± 2.7 km for Filchner-Ronne Ice Shelf (Brunt et al., 2011). Ice shelf at 1 km offshore from the reference GL may not be in full hydrostatic equilibrium for most of the regions.

*Line 163:* The sentence starting on this line is confusing. I think the authors are using the KDTree to get data in proximity to crossover points, and then finding the absolute location using the quadratic functions? This needs to be cleaner/clearer.

The reviewer is correct, we have modified the description in Section 3.2.

*Line 164:* Wrote out and partially describe KDTrees.

Agree and amended.

*Line 168:* What is 'THE valid elevation measurements'?

We have now removed the word 'valid' for clarity.

*Line 173:* "where the elevation exceeded 20 m": It's totally fair to have a cutoff; but suggesting that 20 m is a temporal (accumulation?) change is not the right justification.

We have now changed this threshold to 10 m and changed the sentence into 'we deleted the crossovers where the elevation difference exceeded 10 m to remove other errors such as the geolocation errors of ICESat-2 (Smith et al., 2020)' (line 235-236).

*Line 175:* cite CATS.

Agree and amended.

***Line 177:*** The sentence starting on this line isn't clear to me; I think you used a 40 cm offset for the crossovers based on 20 cm offsets from the repeat-track analysis. But it needs to be cleaner/clearer.

The reviewer is correct, we have now modified the relevant description in line 240-243, please also see our response to Reviewer 1.

***Lines 184 – 191:*** Every reference in this section is missing an 'et al.'; this is common throughout. I'm not looking for all of them.

Agree and amended.

***Line 192:*** "ESA CCI DInSAR GL (ESA Antarctic Ice Sheet Climate Change Initiative, 2017)" Delete the abbreviation part of this.

Amended.

***Line 192:*** How is 'separation' defined? Absolute distance? Distance along the ground tracks?

It was measured as the perpendicular distance from ICESat-2 Point F to DInSAR Point F line segment (line 277-278).

***Line 199:*** For the sentence that starts here, this really isn't a validation; these results just suggest that the point H identified using the repeat-track method is actually on the fully hydrostatic part of the ice shelf; but it doesn't validate that it's the first point of hydrostatic equilibrium, or point H.

We have now removed this sentence.

***Lines 206 – 209:*** I am not sure what this is saying.

We have now removed this section and reexplained the ice plain based on the repeat track analysis for the right beam in track 506 beam pair 1 (Fig. 8), relevant discussion is in line 289-304.

***Lines 206 – 216:*** I do not follow the line of thought here and therefore am not convinced that this rationale and Figure 7 suggest an ice plain.

This is an ice plain based on the example shown in Fig. 8, the detailed explanation is in line 289-304.

***Line 215:*** 'Coupling Line' is not fully defined; the authors should also cite Hugh Corr's work on Pine Island Glacier.

Agree and amended.

***Line 219:*** " ... crossover analysis can provide additional information on the GL location." I don't think that this is true for Larsen C; there isn't the spatial resolution for this to add (or validate) the locations of points F and H. I suggest saving this method (crossover analysis) for a later paper, where the method is more appropriate.

We have now modified the data and the method used in crossover analysis, and updated the distribution of crossovers |dh| on Hearst Island shown in Fig. 9, please also see our response to the second major comment. We think the current results can demonstrate the merit of crossover analysis.

***Lines 225 – 226:*** I do not think that this method provides a validation. What about the pink point above the black F in 8c?

Please see our previous responses.

***Line 231:*** "By assuming that the reference GL used in section 3.1 ..." Cite Depoorter again to remind people what you are referring to.

We have now updated the reference GL used in repeat track analysis by merging the 2015-2016 ESA CCI DInSAR Point F and the Depoorter et al. (2013) GL (Section 3.1.2). Please also see our responses to Reviewer 1.

***Line 234:*** '9.73 km': Why such a large outlier?

The current results show the largest GZ width is 11.62 km which locates inside a concave inlet of a glacier in Churchill Peninsula (Fig. 10). This GZ width could be associated with ice thickness and the geometry of GL in this region, but a detailed investigation of the underlying reasons exceeds the research scope of this study (line 345-351).

***Lines 238 – 240:*** "The aim of this study is to assess the capability and the accuracy of ICESat-2 laser altimetry data for estimating the GZ features of the Antarctic Ice Sheet including flexural point F and hydrostatic point H." I'd remove this; it doesn't offer anything new. But the next sentence does.

Agree and amended.

***Lines 244 – 245:*** "... the new method takes two ground tracks of one beam pair from one repeat cycle as two repeat tracks." I do not understand this sentence.

We have now removed this sentence.

***Lines 253 – 254:*** "...ICESat-1 repeat track analysis which required multiple repeat tracks (normally from at least five repeat cycles) with large temporal differences (Brunt et al. 2010; Brunt et al. 2011)." This statement is not accurate and was not a result in the references provided. There are figures in those papers that use fewer repeats.

Agree and amended. Discussion on the advantage of the capability of our method in identifying GZ based on only two repeat cycles is in line 329-336, please also see our responses to Reviewer 1.

***Line 258:*** (and throughout) As noted in my initial comments, 'precision' is not a great term for this. The close agreement is undoubtedly due to the increased across-track and along-track data density associated with ICESat-2; this is a spatial resolution improvement, not a precision improvement.

We have now removed the word 'precision', please also refer to our response to the third major comment.

***Line 267:*** 'and'; the 2 clauses on either side of this have nothing to do with one another.

Agree and we have modified this sentence into 'proving our method works at a 20 cm scale tidal amplitude in a region with complex relief demonstrates the ability for the generation of GLs for the majority of the Antarctic Ice sheet' in line 300-302.

*Line 272:* The results in this paragraph are a direct repeat of the results from Fricker & Padman 2006, and Brunt et al., 2010, 2011. The authors need to present how they have added to these results.

Agree and amended. We have included a new section 3.1.4 'Filtering and visual validation' to explain how we filtered the wrong GZ identifications due to inaccurate Error function fitting or data deficiency. Please see our detailed response to Reviewer 1.

*Lines 279 – 280:* "provide valuable information about the position of GL in the Antarctic Ice Sheet." I don't think this is accurate. I don't see what these assessments add, with respect to location of the GZ.

Please see our previous responses to the review comments on crossover analysis.

*Line 281:* "it still can show the general location of the GL" I just don't see this.

Please see our previous responses to the review comments on crossover analysis.

*Figure 1:* Consider removing point I or including it in the text with detail. From the figure, it is hard to discern if this is capturing I_m or I_b.

We have now modified Fig.1 to better characterize Point I and included the relevant explanation on Point I in line 33-34 and 48-50. Please also see our previous responses.

*Figure 4b:* I would not repeat the color scheme of 4a; this suggests that different cycles are required for this method.

Agree and amended.

*Figure 5:* Trim the caption. For the legend next to 5b: why are they ordered like this? Why are the purple and olive lines not included? What are the purple and olive lines?? I think they are elevations. It looks like there are 4 lines there (blue and orange??). That is confusing.

Agree and amended.

*Figure 7:* Trim the caption. Some of the language (e.g., defining MAEA again) is repetitive and can be removed. Overall, this figure is confusing and does not convince me that this is an ice plain. Figure 7d is especially confusing.

Agree and amended.

*Figure 8:* 8b shows that the spatial resolution of these crossovers does not add to the results associated with the repeat-track analysis. Suggest removing this.

We have changed this figure into Fig.9 and only showed the distribution of crossovers |dh| on Hearst Island where the ESA CCI DInSAR Point F is not available. The new results can show the approximate location of grounding line, thus we think it's important to keep the crossover analysis in this study. Please also see our previous responses on crossover analysis.

**References cited in authors' response:**

[revised manuscript text omitted]